# An octameric PqiC toroid stabilises the outer-membrane interaction of the PqiABC transport system

Benjamin F Cooper [ID][1], Giedrė Ratkevičiūtė [ID][2], Luke A Clifton [ID][3], Hannah Johnston [ID][4], Rachel Holyfield[4], David J Hardy[4], Simon G Caulton [ID][4], William Chatterton[4], Pooja Sridhar [ID][4], Peter Wotherspoon[4], Gareth W Hughes[5], Stephen CL Hall[3], Andrew L Lovering [ID][4] & Timothy J Knowles [ID][4]✉

## Abstract

**The *E. coli* Paraquat Inducible (Pqi) Pathway is a putative Gram-negative phospholipid transport system. The pathway comprises three components: an integral inner membrane protein (PqiA), a periplasmic spanning MCE family protein (PqiB) and an outer membrane lipoprotein (PqiC). Interactions between all complex components, including stoichiometry, remain uncharacterised; nevertheless, once assembled into their quaternary complex, the trio of Pqi proteins are anticipated to provide a continuous channel between the inner and outer membranes of diderms. Here, we present X-ray structures of both the native and a truncated, soluble construct of the PqiC lipoprotein, providing insight into its biological assembly, and utilise neutron reflectometry to characterise the nature of the PqiB-PqiC-membrane interaction. Finally, we employ phenotypic complementation assays to probe specific PqiC residues, which imply the interaction between PqiB and PqiC is less intimate than previously anticipated.**

**Keywords** Lipid Transport; Outer Membrane Biogenesis; Gram Negative Cell Envelope; Pqi; Envelope Spanning
**Subject Categories** Membranes & Trafficking; Microbiology, Virology & Host Pathogen Interaction; Structural Biology

## Introduction

The Gram-negative cell envelope provides a highly effective barrier against toxic compounds and environmental stresses. Key to its barrier function is the asymmetrical outer membrane (OM) comprising glycerophospholipids (GPLs) and lipopolysaccharide (LPS) within its inner and outer leaflets, respectively. Components of the Gram-negative cell envelope are synthesised within the confines of the inner membrane (IM) where access to biosynthetic enzymes, precursors and cellular energy supplies are plentiful. Consequently, newly synthesised components must traverse the IM and periplasm, often against a concentration gradient, before OM incorporation, all without disrupting the crucial cellular envelope architecture. The amphipathicity of OM components necessitate distinct mechanisms to shield their hydrophobic portions from the aqueous periplasm during trafficking to the OM. Furthermore, these processes must be tightly coordinated to maintain stoichiometric ratios, and once beyond the IM, cellular energy sources such as ATP and the proton motive force (PMF) become unavailable. Whilst the transport pathways responsible for delivering OMPs (Hagan et al, 2011), LPS (Okuda et al, 2016), and lipoproteins (Konovalova and Silhavy, 2015) to the OM have been extensively delineated, the mechanisms facilitating bulk GPL transport remain enigmatic.

MCE proteins (comprising one or more <u>m</u>ammalian <u>c</u>ell <u>e</u>ntry domains) have been implicated in the transport of lipids in mycobacteria (Casali et al, 2006; Casali and Riley, 2007; Chen et al, 2023; de La Paz et al, 2009; Dunphy et al, 2010; Forrellad et al, 2014; Joshi et al, 2006; Kendall et al, 2007; Klepp et al, 2022; Pandey and Sassetti, 2008), plants (Awai et al, 2006; Lu and Benning, 2009) and Gram-negative bacteria (Ekiert et al, 2017; Hughes et al, 2019; Isom et al, 2020, 2017; Malinverni and Silhavy, 2009; Vieni et al, 2022). Indeed, the single MCE domain-containing protein MlaD is a component of the maintenance of the lipid asymmetry (Mla) pathway, which has been extensively characterised both in *E. coli* and *Acinetobacter baumannii*. The Mla system acts to remove mislocalised GPLs from the outer leaflet of the OM, where their presence compromises its barrier function, and return them to the inner membrane (Malinverni and Silhavy, 2009). *E. coli* contains two further MCE domain-containing proteins, LetB (formerly YebT) and PqiB, each residing within unique operons and assembling IM associated, homo-hexamers in vivo (Appendix Fig. S1) (Ekiert et al, 2017; Isom et al, 2020, 2017; Liu et al, 2020;

[1]Sir William Dunn School of Pathology, University of Oxford, OX1 3RE Oxford, UK. [2]Department of Biochemistry, University of Oxford, OX1 3QU Oxford, UK. [3]ISIS Pulsed Neutron & Muon Source, Science and Technology Facilities Council, Rutherford Appleton Laboratory Harwell Oxford Campus, OX11 OQX Didcot, UK. [4]School of Biosciences, University of Birmingham, B15 2TT Birmingham, UK. [5]Institute of Cancer and Genomic Sciences, University of Birmingham, B15 2TT Birmingham, UK. ✉E-mail: t.j.knowles@bham.ac.uk

Nakayama and Zhang-Akiyama, 2017; Vieni et al, 2022). Consequently, these two MCE proteins, and the operons within which they reside, have become of principal interest as putative novel Gram-negative phospholipid transport systems.

PqiB is encoded by the second of three genes residing within the paraquat inducible (*pqi*) operon. The functional homo-hexameric oligomer adopts a syringe-like structure, ~230 Å in length, proposed to span the entire periplasmic space (Ekiert et al, 2017). Each PqiB monomer consists of three consecutive MCE domains and a C-terminal helix which, once assembled into the functional hexamer, contribute to the N-terminal, triple ring-barrel assembly and coiled-coil needle projection, respectively (Appendix Fig. S1) (Ekiert et al, 2017). The PqiB hexamer harbours a central hydrophobic channel running its entire length through which transport is anticipated (Ekiert et al, 2017). The two remaining operonic genes, *pqiA* and *pqiC*, encode an integral IM protein and an OM-associated lipoprotein, respectively (Nakayama and Zhang-Akiyama, 2017). Remarkably, PqiA is predicted to present a unique fold, only displaying sequence homology to LetA (formerly YebS)—the first component of the *letAB/yebST* operon (Jumper et al, 2021) (Appendix Fig. S2). Thus, if PqiA does function as a transporter, it does so either passively, via an as yet unidentified ATPase, or the PMF. PqiC is predicted to anchor within the periplasmic face of the OM and comprises an ABC auxiliary lipoprotein domain (formerly DUF330). Whilst the structure of the *E. coli* protein remains enigmatic, insight can be gained via a 1.45 Å structure of *Enterobacteria cloacae* PqiC (PDB: 6OSX) (Appendix Fig. S2), however, the structure provides little insight into its function nor how it interfaces with PqiB. Indeed, interactions between all complex components, including their final stoichiometry, remain uncharacterised; nevertheless, once assembled into their quaternary complex the trio of Pqi proteins are anticipated to provide a continuous channel between the inner and outer membranes of diderms.

In this study, we focused upon the *E. coli* outer membrane lipoprotein PqiC. We provide X-ray structures of both the native protein and a truncated soluble construct providing insight into its biological assembly and high-resolution side chain information respectively. We utilise neutron reflectometry to investigate the PqiC membrane interaction as well as with a purified PqiAB subcomplex, to our knowledge assembling the first trans-envelope system upon a planar sensor surface. Finally, we employ phenotypic complementation assays to probe specific PqiC residues, which imply the interaction between PqiB and PqiC is less intimate than previously anticipated.

## Results

### PqiC forms an octameric assembly

The oligomeric state of *E. coli* PqiC has remained unclear with structurally homologous proteins having been shown to form both octameric (*Xanthomonas campestris* XCC0632; PDB 2IQI) and dodecameric (*Paraburkholderia phytofirmans* PelC; PDB 5T10) assemblies. Furthermore, the deposited *Enterobacter cloacae* PqiC structure (PDB: 6OSX) displays no clear evidence of a higher order biological assembly. Initially, we overexpressed and purified a construct encoding the entire amino acid sequence of *E. coli* PqiC

(residues 1–187) tagged with a hexa-histidine tag at the C-terminus, referred to as PqiC throughout. This construct is presumably processed and lipidated in the same manner as the native lipoprotein. During purification, PqiC was observed to elute from size exclusion chromatography (SEC) at a volume similar to that of Aldolase (158 kDa) indicative of its assembly into a higher-order multimer (Appendix Fig. S3a). We thus sought to confirm the oligomeric state of *E. coli* PqiC and crystallised the tagged, native protein (Appendix Table S1). The structure, determined at a resolution of 3.2 Å, confirms *E. coli* PqiC as a mixed α/β protein with its two opposite faces comprising a three-stranded antiparallel β-sheet and two α-helices respectively (Fig. 1A–C). Two molecules of PqiC were observed within the asymmetric unit, arranged in a parallel fashion, with the β1-β2 loop, the β6 strand and α2 helix packing against the β7-β9 sheet and β3-β4 loop of the adjacent monomer (Fig. 1A) resulting in a buried surface area of 15,309 Å². The β1-β2 loop region was by far the most poorly resolved, presumably due to its inherent flexibility, thus there was often insufficient density for the explicit positioning of side chains

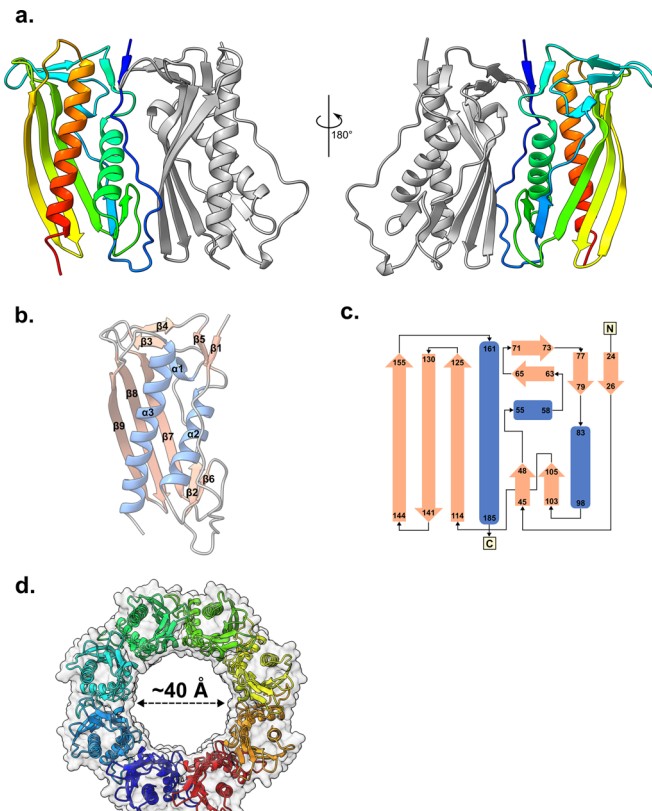

**Figure 1. Structure and biological assembly of PqiC.**

(A) Asymmetric unit of the PqiC crystal structure with its two component chains depicted as cartoons with rainbow and grey colouring respectively. (B) Cartoon representation of a PqiC monomer indicating the secondary structure elements. Coil, helix and strand are indicated in grey, blue and salmon respectively. (C) Topology diagram of PqiC where α-helices and β-sheets are indicated in blue and salmon respectively. (D) Cartoon representation of the PqiC octamer viewed from the membrane plane. Chains coloured in a rainbow gradient, molecular surface depicted in transparent grey and the diameter of the central pore indicated.

throughout this region. Finally, we were unable to model the extreme N-terminus (residues 16–22) of the mature lipoprotein, also presumably due to inherent structural heterogeneity. We utilised the PDBePISA server (Krissinel and Henrick, 2007) to investigate the intermolecular interface in more detail. PISA revealed the interface to be stabilised by 12 intermolecular hydrogen bonds and a single intermolecular salt bridge between D68 and K166 (Appendix Fig. S4, Table S2). Furthermore, the complex formation significance score (CSS) of the interface was reported at 1, indicating the interface to play an essential role in complex formation.

Strikingly, the arrangement of asymmetric units within the unit cell revealed eight copies of PqiC assembled into an octameric toroid structure, ~100 Å in diameter (Fig. 1D). We anticipate this arrangement to represent the PqiC biological assembly, which is consistent with its elution from size exclusion chromatography far earlier than one would expect for a monomeric species of ~20 kDa (Appendix Fig. S3). Furthermore, the homologous structure of *Xanthomonas campestris* XCC0632 (PDB: 2IQI) was solved with an asymmetric unit comprising eight copies in two semi-circular arrangements (Fig. 5H). Once again, the symmetry in the unit cell allowed the formation of complete octameric ring structures analogous to those observed in our PqiC dataset. Finally, the internal diameter of the pore formed by the octameric PqiC arrangement is ~40 Å and thus congruent with the width of the PqiB coiled-coil with which PqiC is anticipated to interact (Fig. 1D).

## Characteristics of the PqiC octamer

Considering the anticipated biological significance of this octameric arrangement, we investigated the characteristics of its molecular surface for indications of structurally and/or functionally critical residues. First, we examined the molecular lipophilicity of the PqiC toroid (Fig. 2A,B), revealing two rings of lipophilic residues navigating the circumference of the primarily hydrophilic central pore (Fig. 2A,E). We defined these rings based on their proximity to the N-terminal lipidated anchor, thus the membrane proximal ring comprises residues Tyr25, Ile73 and Leu78, whilst the membrane distal ring arises from Trp47. The apparent distinction and precise location of these residues imply a possible relation to function. Indeed, if PqiB were to interface with the central pore of PqiC then these residues likely contribute to the stabilisation of such an interaction.

In contrast, the lipophilic residues observed on the outer and membrane distal faces of the PqiC octamer appeared to be predominantly involved in packing interactions to stabilise the quaternary structure. The membrane occluded face was also found to possess lipophilic residues (Tyr64, Val72, Tyr127, Tyr161), which reside predominantly within the centre of this face, flanked by regions of hydrophilicity, suggesting these may interact with the hydrophobic interior of the outer membrane (Fig. 2A,B).

We next considered the coulombic electrostatic potential of the structure (Fig. 2C,D). Once again, the area of primary interest was the lining of the central pore where three distinct regions of negative electrostatic potential were observed (Fig. 2E). In contrast, only a single positively charged residue (Arg88) was found to line the pore (Fig. 2C,D). The top, membrane proximal negatively charged region, arising from Asp55 and Asp84, perfectly bisects the two aforementioned lipophilic regions. The second region, consisting of Glu49, sits adjacent to the membrane distal lipophilic

region hence, at this height, the pore circumference encompasses alternating lipophilic and negatively charged regions (Fig. 2E). Finally, Asp114 sits towards the membrane distal lip of the pore creating a small negatively charged patch almost directly below that of Glu49 (Fig. 2C,D). Consequently, vertical regions of negative charge are present throughout approximately the bottom third of the pore lining (Fig. 2C,E). The role of these charged regions is unclear, however, considering their positioning, it is conceivable that they may form interactions with the coiled-coil of PqiB if the two were to combine as anticipated. Nevertheless, no clear complementary charged regions are apparent towards the C-terminus of the PqiB coiled-coil within AlphaFold multimer predictions (Preprint: Evans et al, 2022) (Appendix Fig. S5) however, the lack of an experimentally determined structure means the register of residues within this region remains ambiguous.

The membrane-occluded face of the PqiC octamer displays distinct regions of electrostatic potential on either side of the lipophilic band identified previously (Fig. 2A,B). Eight regions of positive charge are located around the inner circumference of this face, arising from Lys23 (Fig. 2C,D). In contrast, the outer circumference houses several aspartate residues, (Asp68, Asp128, Asp159 and Asp162), which give rise to regions of negative charge thus the membrane occluded face is electrostatically polarised concentrically (Fig. 2C,D).

Utilising the ConSurf server (Landau et al, 2005), we investigated the evolutionary conservation score (1–9) for each amino acid within PqiC (Appendix Fig. S6). All residues involved in the two lipophilic pore lining bands (Tyr25, Trp47, Ile73 and Leu78), as well as those involved in the negatively charged band composed of residues Asp55 and Asp84 were shown to be highly conserved (Consurf conservation scores ranging from 6 to 9). The enriched conservation of these pore lining residues supports a role in function with Asp55 appearing particularly prominent as a conserved residue (Consurf conservation score of 8). In contrast, Glu49 and Asp114, which give rise to the membrane distal negatively charged band and distal lip negative patches respectively, showed only average conservation (Consurf conservation score of 5).

The charged residues located on the membrane proximal face of PqiC (Lys23, Asp68, Asp128, Asp159 and Asp162) were also found to be conserved (Consurf conservation scores ranging from 6 to 9), potentially supporting their involvement in function. The distinct regions of positive, negative and lipophilic residues upon this face, arising from conserved residues, are likely crucial for the correct orientation and interaction of PqiC with the membrane and require further investigation. Furthermore, AlphaFold multimer predictions (Preprint: Evans et al, 2022) potentially imply the interaction of these charged regions with reciprocal charged regions towards the C-terminus of PqiB (Appendix Fig. S5) however, the somewhat disordered nature and low confidence of the PqiB prediction throughout this region raise doubts about the validity of this interaction (Appendix Fig. S7).

## Structure of the soluble PqiC[17-187] construct

Whilst the lipidated, full length PqiC construct had revealed the octameric assembly of *E. coli* PqiC, the resulting maps were of insufficient resolution to determine accurate side chain information across the entire structure. Consequently, we designed a second N-terminally truncated construct, in which residues 1–16 were replaced with a hexa-histidine tag and TEV protease recognition

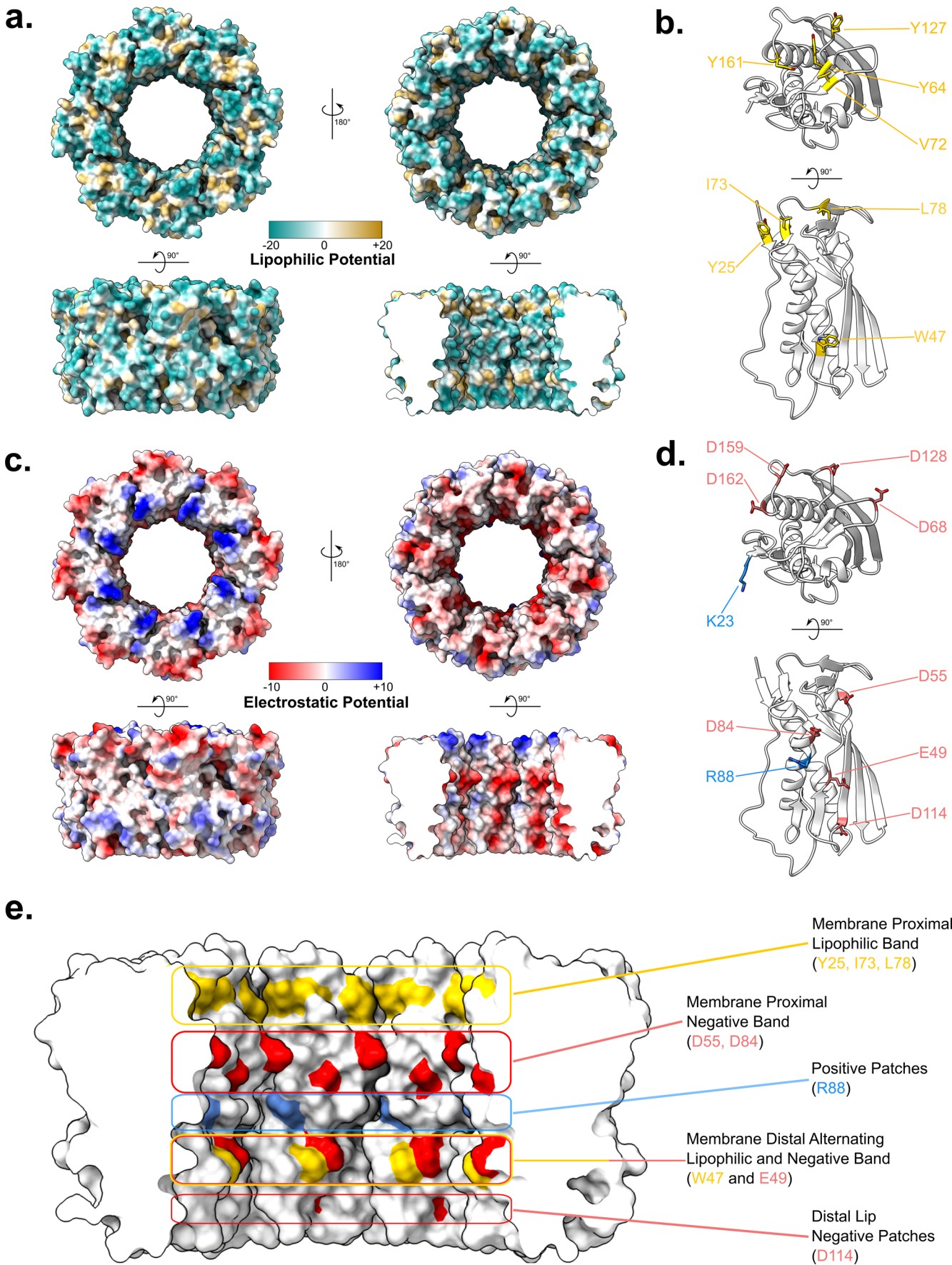

**Figure 2. Lipophilic and coulombic electrostatic potential of the PqiC octamer surface.**

(A) Orthogonal views of the PqiC octamer surface coloured according to molecular lipophilicity with hydrophilic and lipophilic residues indicated in dark cyan and gold, respectively. Views: (left top) from the membrane plane, (right top) from the periplasmic face, (left lower) outer surface, (right lower) inner surface. (B) Cartoon representation of PqiC. Lipophilic residues of interest located within the octameric PqiC pore and membrane occluded face are depicted in gold. (C) Orthogonal views of the PqiC octamer surface coloured according to coulombic electrostatic potential with negative and positive potential indicated in red and blue, respectively. Views: (left top) from the membrane plane, (right top) from the periplasmic face, (left lower) outer surface, (right lower) inner surface. (D) Cartoon representation of PqiC. Positive and negative charged residues of interest located within the octameric PqiC pore and membrane occluded face are depicted in blue and red, respectively. (E) Surface representation of the PqiC pore lining indicating lipophilic (gold) and charged areas (negative - red, positive - blue).

site, referred to as PqiC[17-187], which was expressed cytoplasmically. After TEV cleavage of the N-terminal hexa-histidine tag, the sequence of PqiC[17-187] differed from the native processed PqiC lipoprotein only by the removal of the N-terminally acylated Cysteine (C16). Nevertheless, in contrast to our PqiC construct, we observed PqiC[17-187] to elute from SEC at a volume similar to that of Ribonuclease A (13.7 kDa) (Appendix Fig. S3c). We therefore propose PqiC[17-187] to purify as a monomeric species in contrast to the octameric oligomer of PqiC, implying multimerisation is dependent upon the N-terminal lipidated moiety.

The PqiC[17-187] structure, solved to 2.1 Å (Appendix Table S1), displayed an arrangement unique to that observed in the PqiC structure with three copies present within the asymmetric unit (Fig. 3A). Two of these chains, chains A and B, sit in orientations similar to those of the two corresponding chains in the native PqiC asymmetric unit (Appendix Fig. S8), however, the third chain, Chain C, sits perpendicular and slightly below the plane of the other two, interfacing with the bottom corner of chain A (Fig. 3A). Whilst chains A and B in the PqiC structure sit parallel to one another, in the PqiC[17-187] structure chain A sits marginally higher and is rotated by ~15° relative to chain B (Fig. 3A) (Appendix Fig. S8).

Despite this global geometry change, the monomers from each structure align strongly yielding an RMSD of 0.568 Å across 151 pruned atom pairs and 0.591 Å across all 152 pairs (Fig. 3B). The cores of each structure appear almost identical, with only slight variations observed in the loop regions adjoining secondary structure motifs. Indeed, the large discontinuity of the β1-β2 loop region from the PqiC[17-187] model presents the largest difference between monomers from the two structures (Fig. 3B). Whilst the density corresponding to the first residues of the β1-β2 loop region (residues 27–31) is more clearly resolved than in the full length PqiC structure, the electron density for the extremity of the loop (residues 32–44) was mostly absent from the 2mFo – DFc map contoured at 1 sigma preventing modelling of this region (Fig. 3C). Despite this, the electron density corresponding with the core of the protein was a considerable improvement over that of the full length PqiC structure with side chain information clearly defined in both β-sheet and α-helical regions (Fig. 3C). Overall, our PqiC[17-187] structure provided us with high-resolution side chain information throughout PqiC, with the exception of the β1-β2 loop region, confirming the conformation of key residues of interest lining the octameric pore.

## PqiC embeds into the membrane lipid headgroups

As PqiC is predicted to be an outer membrane-associated lipoprotein, we next sought to understand how PqiC might interact with a phospholipid bilayer. Using a combination of quartz crystal microbalance with dissipation monitoring (QCM-D) and neutron

reflectometry (NR) we examined the orientation and penetration of PqiC on a silicon-supported DMPC bilayer. Although not a constituent of native *E. coli* membranes, DMPC was utilised due to its close approximation to phosphatidylethanolamine (PE), the major constituent of the bacterial membrane, whilst also being amenable to planar bilayer formation and available in deuterated forms. These characteristics yielded a system highly amenable to supported bilayer deposition whilst also providing a realistic structural analogue of the inner leaflet of the *E. coli* outer membrane. QCM-D enabled real-time monitoring of the deposition process via mass changes on the sensor surface whilst NR, using an isotopic contrast variation approach, allowed structural resolution of the protein, lipid and water distributions within the interfacial assembly.

QCM-D measurements indicated that, following equilibration in buffer, the addition of PqiC-DMPC proteoliposomes led to a large decrease in resonant frequency (~50 Hz) and a concomitant increase in dissipation (~3 ppm), characteristic of the deposition of a viscoelastic material, commonly associated with vesicle adsorption, onto the silicon surface (Fig. 4A). An exchange of buffer into $H_2O$, to promote osmotic shock was performed, with subsequent return to buffer. This resulted in an overall resonance frequency decrease of ~49 Hz for the lipid deposition process. This is larger than one would expect for deposition of a DMPC bilayer alone (~27 Hz) (Appendix Fig. S9a) and is consistent with the additional PqiC within the system, however, clear evidence of vesicle rupture was not observed as characterised by a sudden dip in signal before levelling out (Fig. 4A).

To elucidate the interfacial structure created, we utilised NR. NR is uniquely able to probe the interfacial structure normal to the surface with ångström scale precision and is ideally suited to probe protein position and distribution both on and within the membrane. To provide the necessary scattering length density contrast between protein and lipid we used deuterated DMPC in subsequent experiments. Results confirmed that deposition of PqiC containing deuterated DMPC proteoliposomes and subsequent osmotic shock led to the formation of a planar bilayer supported on a silicon surface with a protein layer present solely on the leaflet distal from the silicon support (Fig. 4). This protein layer had a thickness of ~50 Å, consistent with a vertical PqiC, orientated with its helices perpendicular to the bilayer (Fig. 4B–D). The position of the protein layer suggests its intercalation into the lipid headgroup region. Whilst this may result from lipidation of Cys16, as a native PqiC construct was used, the volume fraction of protein within the lipid headgroup region suggests the core domain itself is likely intercalated (Fig. 4D) however, further penetration of the protein into the lipid acyl tails was not observed.

Collectively, our NR results are consistent with our octameric PqiC structure and suggest that PqiC sits upon the surface of the

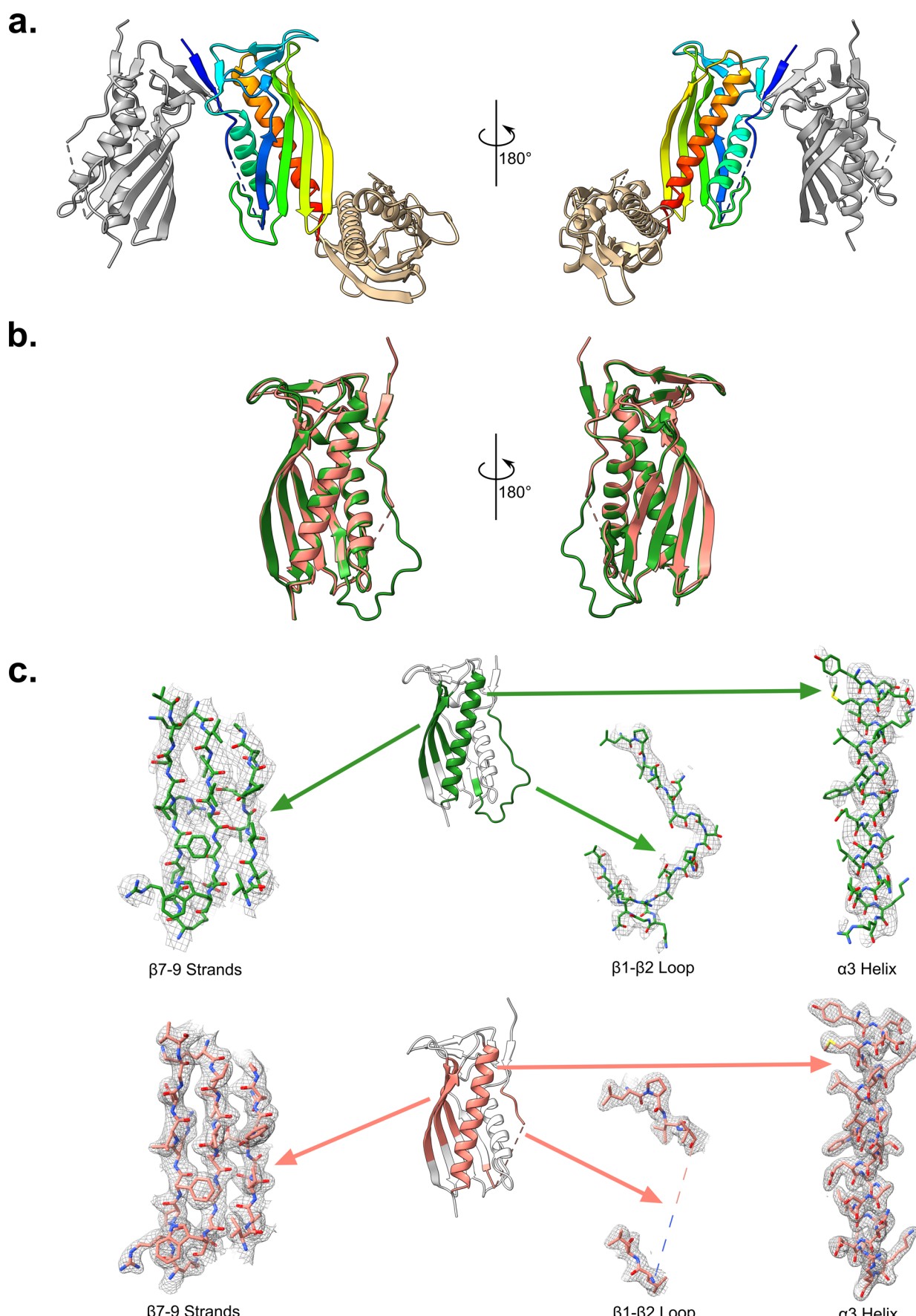

β7-9 Strands

β1-β2 Loop

α3 Helix

β7-9 Strands

β1-β2 Loop

α3 Helix

**Figure 3. Comparison of the PqiC and PqiC$^{17\text{-}187}$ structures.**

(A) Cartoon representation of the PqiC$^{17\text{-}187}$ structure asymmetric unit with chains A, B and C coloured in rainbow gradient, grey and tan respectively. (B) Alignment of chain A from the PqiC (green) and PqiC$^{17\text{-}187}$ (salmon) structures highlighting the similarity in their overall architecture. (C) Final refined electron density map for the PqiC and PqiC$^{17\text{-}187}$ structures is displayed for three key regions of the protein—the β1-β2 loop, the α3 helix and the β7-β9 antiparallel strands. The 2mFo – DFc map (grey mesh) is contoured at 1 sigma and protein displayed as ball and stick representation coloured green (PqiC) and salmon (PqiC$^{17\text{-}187}$). The position of these features in the protein is indicated by the coloured regions on the cartoon representations.

bilayer, embedding into the lipid headgroups, with the bulk of its torus residing within the periplasm (Fig. 4C,D).

## PqiC tethers PqiAB to the outer membrane

Considering the anticipated interaction between PqiC and PqiB upon assembly of the entire PqiABC complex, we investigated whether the surface-tethered PqiC octamer we observed in our NR data was able to receive a purified PqiAB subcomplex (Appendix Fig. S10), confirming its biological relevance. QCM-D experiments showed a large decrease in frequency (93 Hz— at point of exchange back into buffer), upon the addition of PqiAB containing proteoliposomes to a PqiC-membrane surface (Fig. 4E). No loss was evident following subsequent exchange back into the buffer alone, highly indicative that a stable deposition of PqiAB on the surface had occurred. The large increase in dissipation further confirms system formation, and is suggestive of the deposition of a highly visco-elastic material, entirely consistent with the binding of PqiAB proteo-liposomes. In the absence of PqiC, deposition was significantly curtailed (<20 Hz), with any deposition likely non-specific as buffer washing was sufficient to remove it (Appendix Fig. S9a). Reverse experiments, depositing a PqiC-containing bilayer and subsequent DMPC addition showed no changes, confirming the interaction between PqiAB and PqiC is down purely to protein:protein interactions, consistent with PqiAB tethering to the PqiC toroid and forming a stable complex (Appendix Fig. S9b).

To understand the nature of the PqiB-PqiC interaction, we utilised NR to probe the interfacial structure following PqiAB-proteoliposome addition (Fig. 4F–H). The NR reflectivity curves were feature rich enabling accurate model fitting of the data (Fig. 4F). Results confirmed ordered deposition of PqiAB-proteoliposome material onto the surface with no loss of PqiC or the supported bilayer. Best fits were achieved by modelling a 230 Å proteinaceous element extending out from PqiC on the surface, anchoring a diffuse mixed protein/lipid layer (Fig. 4G). These results are entirely consistent with the predicted length of PqiB, thus we anticipate tethering of the PqiA containing proteoliposome to the PqiC-DMPC bilayer via this proteinaceous moiety (Fig. 4H). The 230 Å separation between our two modelled DMPC regions is comparable with the observed diameter of the *E. coli* periplasm suggesting that the trans-envelope Pqi architecture observed here may indeed reflect that adopted in vivo. To our knowledge, this is the first example of the assembly of a trans-envelope system upon a planar sensor surface.

## PqiABC is tolerant to mutations within the PqiC octameric pore

Whilst our QCM-D and NR data had confirmed an interaction between the octameric PqiC toroid and hexameric PqiB, the nature of this interaction remained unclear. Indeed, considering the data available to us, the most likely assembly involved the insertion of the PqiB coiled-coil into the pore of the PqiC octamer. Therefore, guided

by our X-ray structures, lipophilicity, electrostatic and conservation analysis, we identified four residues of interest, lining the PqiC pore, which warranted further analysis (Fig. 5A). Leu78, which forms part of the membrane proximal lipophilic band; Arg88, responsible for the only region of positive charge within the PqiC pore; and finally Asp55 and Asp84, both of which form the major negatively charged band towards the centre of the PqiC pore. Whilst Consurf conservation analysis revealed both Asp55 and Asp84 to be conserved, Asp55 was indicated as one of the most highly conserved PqiC residues. Furthermore, our PqiC structures indicate the side chains of all four of these residues extend towards the centre of the pore suggesting involvement in complex function or assembly (Fig. 5B).

To probe the importance of the selected PqiC residues we performed established phenotypic complementation assays to assess the ability of plasmid-encoded copies of the *pqiABC* operon (WT or mutated) to rescue the growth of an *E. coli* Δ*pqiABC* strain in the presence of the detergent lauryl sulfobetaine (LSB) (Isom, 2017). All complementation assays utilised the same three controls: 1 – the WT *E. coli* BW25113 (hereafter referred to as WT), 2 – our own *E. coli* BW25113 Δ*pqiABC* knockout strain (hereafter referred to as ΔABC), and 3 – the ΔABC strain transformed with a plasmid harbouring the *pqiABC* operon, hexahistidine tagged upon the C-terminus of PqiC (hereafter referred to as ΔABC + ABC).

Interestingly, the ΔABC + ABC strain showed improved growth on LSB, compared to the WT strain, presumably a result of increased PqiABC levels from plasmid-encoded expression providing increased tolerance to LSB. In the presence of LSB, the PqiC L78D, R88M, D55A, D84A and D84K single mutants all displayed similar growth to the WT strain, but less than the ΔABC + ABC strain (Fig. 5C) (Appendix Fig. S11), whilst the D55K mutant displayed the poorest growth, being less than that of the WT strain and substantially lower than that of the ΔABC + ABC strain (Fig. 5C) (Appendix Fig. S11). Western blotting against the hexahistidine tag on the C-terminus of PqiC revealed the levels of expressed protein within all the single mutants to be comparable to that of the ΔABC + ABC strain (Appendix Fig. S12). The reduced growth of all single mutants in the presence of LSB compared to the ΔABC + ABC strain, despite comparable protein expression, implies these mutations do provoke minor functional disruption, in particular D55K, though these disruptions are minimal and not sufficient to completely abolish Pqi functionality.

Whilst none of our single mutations were able to completely abolish Pqi function, the charge reversal of D55K appeared to induce the greatest disruption. Consequently, we considered the possibility that removal or reversal of a larger charged region within the PqiC pore may elicit more significant effects. We therefore created a series of double mutants focusing on the membrane proximal negative band arising from D55 and D84 (Fig. 2). Simultaneous substitutions of both residues to alanine (D55AD84A) and charge reversal (D55KD84K) yielded similar growth to the WT strain in the presence of LSB (Fig. 5D). To our surprise, despite their apparent ability to

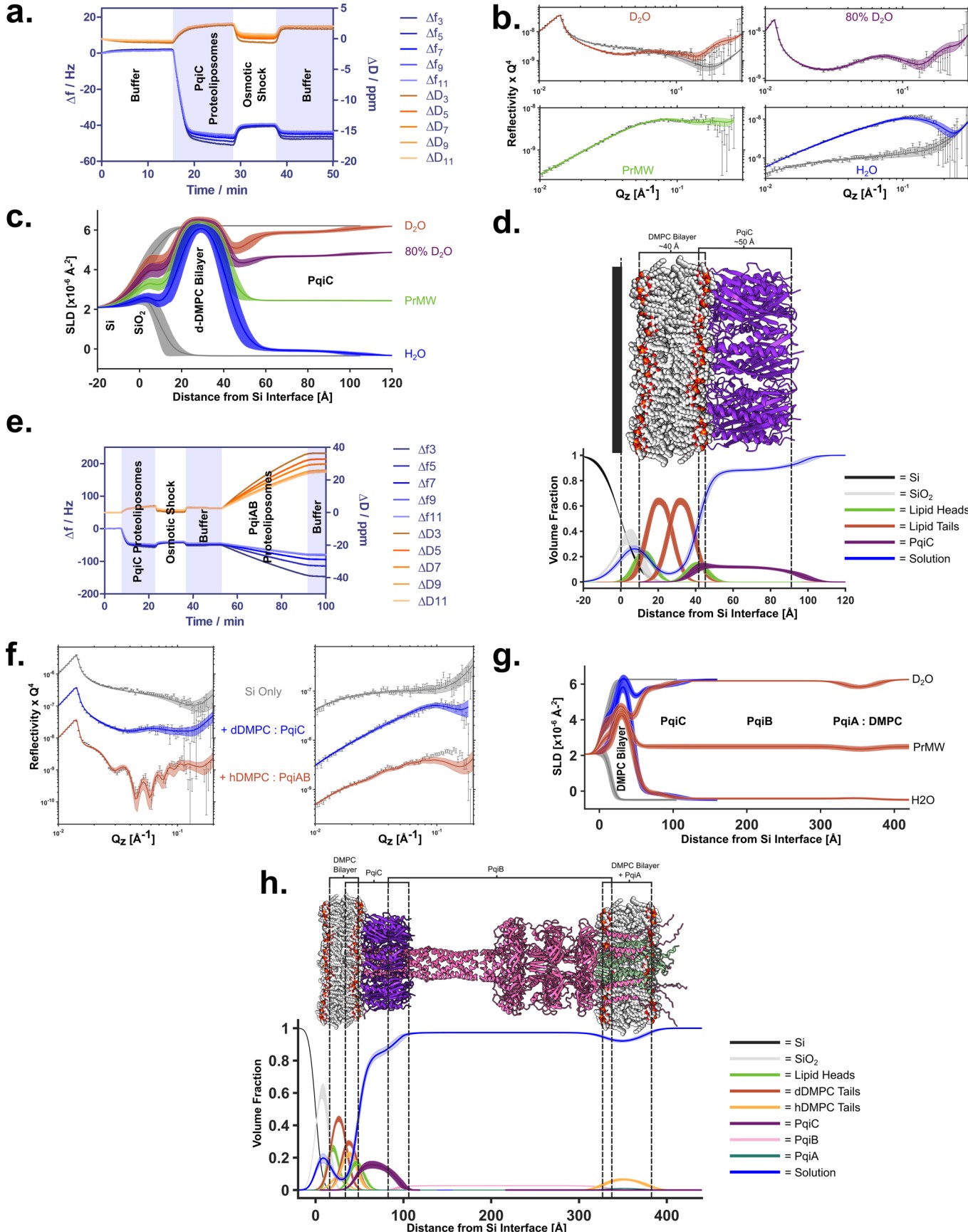

**Figure 4. QCM-D and NR suggest the embedding of PqiC into the outer membrane inner leaflet.**

(A) QCM-D measurements indicating the deposition of PqiC-DMPC proteoliposomes on a silicon surface and their subsequent transition to a PqiC containing DMPC bilayer. (B) Neutron reflectometry profiles (error bars) and model data fits (lines) for a silicon surface before (grey lines) and after the deposition of a deuterated DMPC bilayer with bound PqiC under four buffer solution isotopic contrast conditions being $D_2O$ (red), 80% $D_2O$ (purple), Protein matched water (PrMW, green) and $H_2O$ (blue). Line widths in the NR data fits represent the 65% confidence interval of the range of acceptable fits determined from Mont Carlo Markov chain (MCMC) error analysis. This experiment was repeated in triplicate. (C) The scattering length density distance profile for the interfacial components determined from the model data fits. Line widths for the model data fits represent the range of accepted fits from the Markov chain, which resides within a 65% confidence of the parameter posterior distributions from Bayesian error estimation of the fits. (D) Model of the PqiC octamer - DMPC bilayer interaction aligning with the NR volume fraction fits. Line widths on the volume fraction profiles represent the 65% confidence intervals from the interfacial structural parameters determined from the same process. (E) QCM-D measurements showing reproducible deposition of a PqiC planar bilayer (as in A) but with subsequent addition of PqiAB-proteoliposomes and an observed large frequency decrease, consistent with the formation of a stable PqiB-PqiC interaction. (F) Neutron reflectometry profiles (error bars) and model data fits (lines) from the sequential self assembly of the Pqi complex between two model membranes on a silicon surface in both $D_2O$ and $H_2O$ solution contrast conditions. NR data and fits from the bare silicon surface (grey line), the same surface with a dDMPC bilayer with PqiC deposited (blue line) and, finally, PqiAB and hDMPC added to this (red line) are shown. Line widths in the NR data fits represent the 65% confidence interval of the range of acceptable fits determined from Mont Carlo Markov chain (MCMC) error analysis. This experiment was repeated in duplicate. (G) The neutron scattering length density profiles from each stage of the assembly process are shown with the same representative colour scheme. Line widths on the model data fits represent the range of accepted fits from the Markov chain which reside within a 65% confidence of the parameter posterior distributions from Bayesian error estimation of the fits. (H) Model of the sequentially assembled PqiABC complex aligning with the NR component volume fraction vs distance profile for all the resolved structural components across the interface for the fully assembled PqiABC sample between two model bilayers. Line widths on the volume fraction profiles present the 65% confidence intervals from the interfacial structural parameters determined from the same process.

complement the ΔABC strain, western blotting revealed the level of protein expressed by both the double mutants to be substantially lower than that of the ΔABC + ABC and single mutant Pqi constructs (Appendix Fig. S12) implying only a few molecules of PqiC are required for functionality.

Following our observation that the PqiC double mutants were able to complement the ΔABC strain, regardless of their low expression, we considered the possibility that PqiC itself was superfluous to Pqi function. To investigate this possibility, we tested the ability of two plasmids comprising the *pqiAB* genes, tagged at the N-terminus of PqiA (His-AB) and the C-terminus of PqiB (AB-His) respectively, to complement the growth of the ΔABC strain in the presence of LSB. Whilst our previous complementation studies had demonstrated that the hexa-histidine tag on the C-terminus of PqiC did not impact Pqi function, we were unsure if tagging either the PqiA N-terminus or PqiB C-terminus would impact function. Consequently, we tested the two PqiAB constructs alongside two PqiABC constructs, the construct tagged at the C-terminus of PqiC used thus far (ABC) and a new construct tagged at the N-terminus of PqiA (His-ABC).

When grown on LSB, both the N- and C-terminally tagged PqiAB plasmid-containing strains displayed growth comparable to the ΔABC strain, indicating the complete abolition of Pqi function (Fig. 5D) (Appendix Fig. S11). In contrast both the N- and C-terminally tagged PqiABC constructs displayed comparable growth to each other, which was once again slightly higher than that of the WT strain, implying neither tag position to be detrimental (Fig. 5D) (Appendix Fig. S11). Consequently, we conclude that the inability of the His-AB strain to complement, whilst the His-ABC strain remains viable on LSB, demonstrates the requirement of PqiC for Pqi function. Overall, these results imply the necessity of PqiC for PqiABC complex function but suggest the central pore, if it does indeed couple with the coiled-coil of PqiB, is highly tolerant to mutagenesis suggesting any interactions are likely non-specific.

## Discussion

In this study we have presented the octameric X-ray structure of PqiC which we believe to be biologically relevant based on the following findings: 1- the SEC elution profile of PqiC is consistent with it forming an oligomeric complex (Appendix Fig. S3) and 2 - the internal diameter of the octameric pore is congruous with the width of the PqiB coiled-coil with which PqiC is anticipated to interact (Fig. 1D).

To further verify our assumption of this octameric architecture we utilised the Dali server (Holm, 2020), run against a PqiC monomer, to identify homologous structures deposited to the PDB. Dali returned six hits with an RMSD below 2.5 Å; *Enterobacter cloacae* PqiC (PDB: 6OSX), *Xanthomonas campestris* XCC0632 (PDB: 2IQI), *Paraburkholderia phytofirmans* PelC (PDB: 5T10), *Geobacter metallireducens* PelC (PDB: 5T0Z), *Acinetobacter baumannii* LptE (PDB: 5TSE) and *Shewanella oneidensis* LptE (PDB: 2R76) (Fig. 6A–F). In support of our observed octameric PqiC structure, the unit cell of the *Xanthomonas campestris* XCC0632 structure contains eight copies within the asymmetric unit, which form two semi-circular arrangements, each comprising four monomers (Fig. 6G). Viewing symmetry copies reveals that the crystal is composed of octameric rings analogous to those present within our PqiC structure reinforcing the potential biological relevance of this arrangement. Curiously the operon to which *Xanthomonas campestris* XCC0632 belongs also houses homologues of the *mlaFE* ABC-transport genes in addition to an MCE domain containing protein. Such architectures have been observed elsewhere, including *Legionella pneumophila* (Nakayama and Zhang-Akiyama, 2017), implying that, in these species, retrograde transport may proceed through a tunnel-like Mla system—a notion requiring further investigation. Finally, the deposited *Paraburkholderia phytofirmans* PelC structure also forms a ring, albeit a dodecamer (Fig. 6H), however it must be noted that the structure of the PelC monomer is distinctly different to that of PqiC, with the largest differences being observed at the N-terminus (Fig. 6C). Although not conclusive, the apparent propensity of structurally homologous proteins to adopt ring-like conformations during crystallisation reinforces our belief of the existence of such architectures in vivo.

We also provide the structure of a soluble PqiC[17-187] construct which differs from the mature processed PqiC lipoprotein only by the removal of the N-terminally lipidated cysteine (C16). Despite yielding a near identical monomer conformation, the PqiC[17-187]

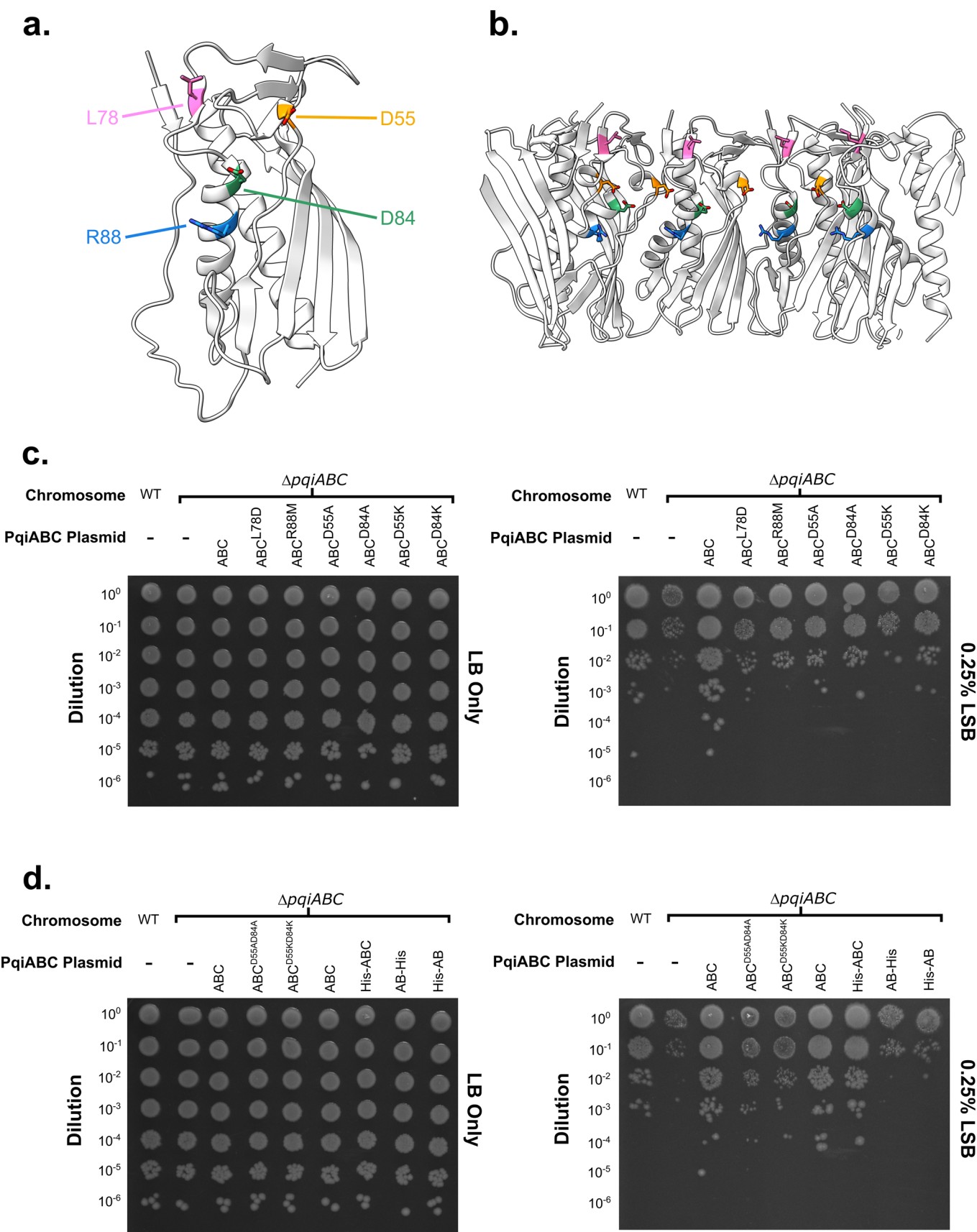

 **Figure 5.** **PqiC is required for Pqi function and resistant to mutations within its pore.**

(A) Structure of PqiC with the residues mutated during complementation assays indicated. (B) Cartoon representation of the PqiC pore lining indicating the position of mutated residues in the context of the PqiC octamer. (C) Phenotypic complementation assay assessing the importance of single residues lining the PqiC pore. (D) Phenotypic complementation assay assessing double mutations within the PqiC pore and the requirement of PqiC for Pqi function. Source data are available online for this figure.

construct produced a dramatically different asymmetric unit with three copies as opposed to the two present in the native PqiC structure (Figs. 1A and 3A). We propose this differing arrangement results directly from the soluble or membrane/detergent-associated nature of the two constructs. As a lipoprotein, native PqiC sits tethered to the membrane with the possibility of interacting directly with the membrane surface itself, a notion predicted by molecular dynamics simulations (Rao et al, 2020) and confirmed through our own NR investigation (Fig. 4B–D). For the detergent solubilised native PqiC, the movement of a single PqiC monomer is severely restricted as its translation vertically is limited by both the presence of the detergent micelle and the length of its acylated, N-terminal tether. Lateral translation is also restricted by the additional monomers present within the octameric assembly and the interactions stabilising this quaternary state. In contrast, the PqiC[17-187] construct was observed to purify as a monomeric species (Appendix Fig. S3b), and is presumably unable to adopt the same octameric assembly as PqiC. Consequently, PqiC[17-187] monomers are able to adopt a conformation more favourable for crystal packing thus, the molecules within the unit cell no longer assemble an octameric structure. This observation potentially explains the lack of an octameric assembly in the deposited structure of *Enterobacter cloacae* PqiC (PDB: 6OSX), which crystalised in the same P 6₁ space group as our PqiC[17-187] construct.

It has been speculated that PqiC interacts with PqiB based on the colocalisation of the two genes within a single operon. Here, we confirm this interaction and provide insight by demonstrating that octameric PqiC sits embedded within the headgroup region of the membrane coordinating with the PqiB coiled-coil. Although the precise nature of their engagement remains enigmatic, the diameters of the PqiB coiled-coil projection and octameric PqiC pore are consistent with these being the likely interfaces (Fig. 7A,B).

The distance between the two reconstituted membranes in our PqiABC NR system was calculated to be 28.1 ± 1.4 nm. This value is congruous with the upper limit of the 21–27 nm range for the measured width of the periplasm in vivo (Mandela et al, 2022; Matias et al, 2003). This is much larger than the 200 Å estimate of the periplasmic span during assembly of the translocation and assembly module upon planar surfaces using both QCM-D and NR (Selkrig et al, 2015; Shen et al, 2014). Here the system comprised the Omp85 outer membrane protein TamA and the AsmA-like protein TamB. However, this work differs from our own in one fundamental aspect. Here, the use of a TamB construct lacking its N-terminal TM region prevented the formation of a second distal membrane. In contrast, the use of native PqiB and PqiA constructs within our own studies enabled the sequential assembly of the complete trans-envelope Pqi complex with a double membrane system. Previous cryo-EM studies have reported the periplasmic portion of PqiB to be ~230 Å in length (Ekiert et al, 2017), whilst our own NR studies reveal PqiC to sit ~50 Å proud of the outer-membrane surface. Consequently, the observed distance may be

accounted for simply by the stacking of these two components and thus insertion of the PqiB coiled-coil into the PqiC toroid is not required. Nevertheless, we anticipate the interaction between PqiB and PqiC to require the PqiB coiled-coil to locate into the centre of the PqiC octamer, a notion further supported by the AlphaFold multimer prediction of the PqiABC complex (Preprint: Evans et al, 2022) (Fig. 7) (Appendix Figs. S5, 7). Thus, assuming the insertion of PqiB within PqiC, the marginally larger than anticipated intermembrane distance measured may simply reflect the inherent low resolution in vitro nature of our NR study. It must also be noted that our NR experiments were conducted with the flow cell located below the silicon substrate, thus the assembled system, with the PqiAB-liposome tethered, was suspended beneath the substrate and would therefore be subject to the effects of gravity.

The structure of PqiC also highlights an interesting symmetry mismatch. PqiB exhibits the canonical hexameric MCE protein architecture (Ekiert et al, 2017) (Appendix Fig. S1a), whilst our results show PqiC to be octameric in nature. Interactions between MCE proteins and components of differing symmetries have been well documented, with several structures revealing a hexameric MCE protein assembly atop dimeric ABC transporter machinery (Chen et al, 2023; Chi et al, 2020; Coudray et al, 2020; Mann et al, 2021). Symmetry mismatch has also been observed in other trans-envelope systems including both the Type 2 (T2SS) and Type 3 (T3SS) Secretion Systems. Here, 15:12 and 15:24 stoichiometries have been observed between the C-modules and their IM partners, respectively (Chernyatina and Low, 2019; Hu et al, 2018). Exactly how this symmetry mismatch might be overcome remains unclear, but there is mounting evidence to suggest the overall stoichiometry of the secretin may fluctuate to be compatible with the stoichiometry of its IM partner (Barbat et al, 2023; Hay et al, 2017). In principle the same could occur for PqiC, however, this seems somewhat less likely due to the coinciding diameters of the hexameric PqiB coiled-coil and octameric PqiC pore.

Interestingly, despite being guided by our X-ray structures, none of our PqiC pore mutations appeared to majorly impede Pqi function during phenotypic complementation assays. Nevertheless, our inability to complement the ΔABC strain with a PqiAB expressing plasmid implies PqiC to be indispensable for correct Pqi function. The ability to tolerate charge reversal mutations within the PqiC pore suggests a lack of intimate interactions between PqiB and PqiC. This observation, together with the apparent PqiB-PqiC symmetry mismatch and our NR data confirming PqiC sits upon the membrane, suggests a scenario in which PqiC might act purely to tether the PqiC coiled-coil without forming specific associations, akin to a bearing (Fig. 7). The single existing cryo-EM structure of a soluble PqiB construct, solved in isolation, (PDB: 5UVN) displays a straight coiled-coil region, owed to the application of C6 symmetry, however indications of its flexibility are evident within the associated publication (Ekiert et al, 2017). Both initial negative stain EM and subsequent CryoEM 2D classes clearly indicate an

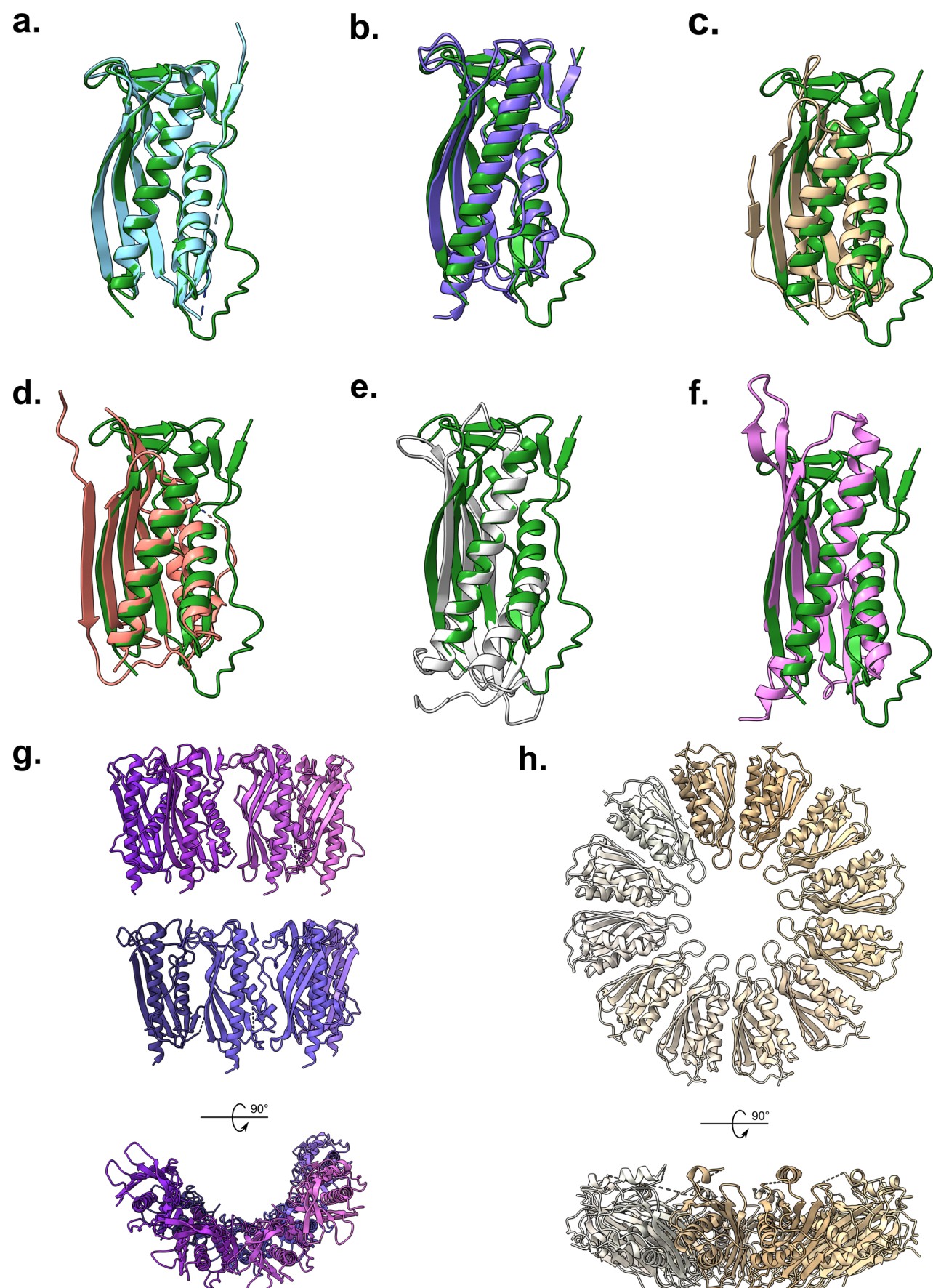

◄ **Figure 6. Structural homologues of PqiC identified by the Dali Server.**

Aligned cartoon representations of PqiC chain A (green) and single chains from; (A) *Enterobacter cloacae* PqiC (PDB: 6OSX), (B) *Xanthomonas campestris* XCC0632 (PDB: 2IQI), (C) *Paraburkholderia phytofirmans* PelC (PDB: 5T1O), (D) *Geobacter metallireducens* PelC (PDB: 5TOZ), (E) *Acinetobacter baumannii* LptE (PDB: 5TSE) and (F) *Shewanella oneidensis* LptE (PDB: 2R76). (G) Cartoon representation of the *Xanthomonas campestris* XCC0632 asymmetric unit, encompassing two stacked semi-circular arrangements, which form octameric rings throughout the crystal (PDB: 2IQI). (H) Cartoon representation of the anticipated *Paraburkholderia phytofirmans* PelC dodecameric ring biological assembly (PDB: 5T1O).

array of PqiB coiled-coil poses, displaying a radial sweeping motion. Indeed, the authors themselves state this flexibility limited their ability to model the coiled-coil region accurately (Ekiert et al, 2017).

We therefore propose PqiC acts as a collar, anchoring the C-terminal moiety of PqiB to the OM limiting its flexibility, whilst potentially allowing its movement/rotation within PqiC (Fig. 7C,D). This model also provides rationale for the apparent essentiality of PqiC, as without the PqiC octamer, we propose that the C-terminus of PqiB is unable to form its essential interaction with the OM. Nevertheless, it remains unclear if PqiC is required to simply stabilise a direct interaction between PqiB and the OM or resides upon the membrane acting as a coupler. Indeed, several fundamental uncertainties persist regarding the structure and function of the *E. coli* Pqi complex. Despite a strong consensus, evidence implicating Pqi as a lipid transport system is still speculative, furthermore, its transport mechanism, regulation and directionality all remain enigmatic. These principal uncertainties provide obvious aims for future studies aiming to characterise the Pqi complex.

## Methods

### Expression and purification of PqiC

DNA corresponding to *E. coli* PqiC residues 1–186 was chemically synthesised (Genscript) and cloned into a pET 22b plasmid, between the Nde1 and Xho1 restriction sites, to contain a C-terminal hexa-histidine tag, which we refer to as PqiC throughout (Appendix Table S5). The resulting plasmid was transformed into the *E. coli* C43 (DE3) cell line (Miroux and Walker, 1996) (Appendix Table S6) for expression. Cultures were grown at 37 °C in Miller's lysogeny broth (Melford), supplemented with 100 μg/mL ampicillin, until an $OD_{600}$ of 0.4 at which point the temperature was reduced to 18 °C. Upon obtaining an $OD_{600}$ of 0.6, expression was induced via the addition of 1 mM Isopropyl β-d-1-thiogalactopyranoside (IPTG) and proceeded overnight at 18 °C with continual shaking (180 rpm). Cells were harvested via centrifugation at 6000 RCF for 15 minutes. Cell pellets were resuspended in an appropriate volume of 20 mM Bis-Tris Propane, 500 mM NaCl; pH 8.5 supplemented with cOmplete™ EDTA-free protease inhibitor cocktail tablets (Roche). Cells were lysed via four passes through an Emulsiflex-C3 cell disruptor (Avestin) before cell debris was removed via centrifugation at 10,000 RCF for 10 min. The supernatant was decanted and spun at 165,000 RCF for 1 h to pellet the membrane fraction. Membrane pellets were resuspended in 20 mM Bis-Tris Propane, 500 mM NaCl; 1% w/v n-dodecyl-β-D-maltoside (DDM); pH 8.5 at a ratio of 1 ml per 40 mg of wet membrane mass and solubilised for 3 h at 4 °C. Insoluble material was removed via centrifugation at 75,000 RCF for 30 min and the resulting supernatant filtered through a 0.45 μm syringe filter

(Sartorius) before clarified lysate was incubated overnight at 4 °C with 1 mL of Ni-NTA Agarose (Qiagen) pre-equilibrated with 20 mM Bis-Tris Propane, 500 mM NaCl, 50 mM Imidazole, 0.03% w/v n-dodecyl-β-D-maltoside; pH 8.5. The incubated supernatant was transferred to a gravity column, washed with 20 mM Bis-Tris Propane, 500 mM NaCl, 50 mM Imidazole, 0.03% w/v n-dodecyl-β-D-maltoside; pH 8.5 before elution in 20 mM Bis-Tris Propane, 500 mM NaCl, 500 mM Imidazole, 0.03% w/v n-dodecyl-β-D-maltoside; pH 8.5. Fractions containing PqiC were pooled, concentrated and further purified via a Superdex 200 16/60 column (Cytiva) equilibrated with 20 mM Tris, 150 mM NaCl, 0.03% w/v n-dodecyl-β-D-maltoside; pH 8.5 (Appendix Fig. S3a,b).

### Expression and purification of PqiC$^{17-186}$

DNA corresponding to PqiC residues 17–186, with the addition of an N-terminal hexa-histidine tag and TEV protease recognition site, was chemically synthesised (Genscript) and cloned into the pET 26b plasmid, which we refer to as PqiC$^{17-186}$ throughout (Appendix Table S5). The resulting plasmid was transformed into the *E. coli* C43 (DE3) cell line (Miroux and Walker, 1996) (Appendix Table S6) for expression. Cultures were grown overnight at 37 °C in Miller's lysogeny broth (Melford), supplemented with 30 μg/mL kanamycin, until an $OD_{600}$ of 0.4 at which point the temperature was reduced to 18 °C. Upon obtaining an $OD_{600}$ of 0.6, expression was induced via the addition of 1 mM IPTG and proceeded overnight at 18 °C with continual shaking (180 rpm). Cells were harvested via centrifugation at 6000 RCF for 15 min. Cell pellets were resuspended in an appropriate volume of 20 mM Tris, 500 mM NaCl; pH 8.5 supplemented with cOmplete™ EDTA-free protease inhibitor cocktail tablets (Roche). Cells were lysed via four passes through an Emulsiflex-C3 cell disruptor (Avestin) before cell debris was removed via centrifugation at 50,000 RCF for 1 h. The resulting supernatant was filtered through a 1.2 μm syringe filter (Sartorius) and the clarified lysate bound overnight to a 5 ml HisTrap HP column (Cytiva) pre-equilibrated in 20 mM Tris, 500 mM NaCl, 50 mM Imidazole; pH 8.5. The column was then washed with 20 mM Tris, 500 mM NaCl, 50 mM Imidazole; pH 8.5 before elution in 20 mM Tris, 500 mM NaCl, 500 mM Imidazole; pH 8.5. Fractions containing PqiC$^{17-186}$ were pooled, concentrated and further purified via a Superdex 200 26/600 column (Cytiva) equilibrated in 20 mM Tris, 150 mM NaCl; pH 8.5. Fractions containing PqiC$^{17-187}$ were pooled and incubated with TEV-protease overnight. Cleaved protein was isolated from the hexa-histidine tag by reflowing through a 5 mL HisTrap HP column (Cytiva) equilibrated in 20 mM Tris, 150 mM NaCl; pH 8.5 (Appendix Fig. S3c,d).

### Crystallisation and structure determination of PqiC

Size exclusion fractions of PqiC were concentrated to between 0.5 and 15 mg/mL in a 10 kDa MWCO Amicon Ultra centrifugal filter

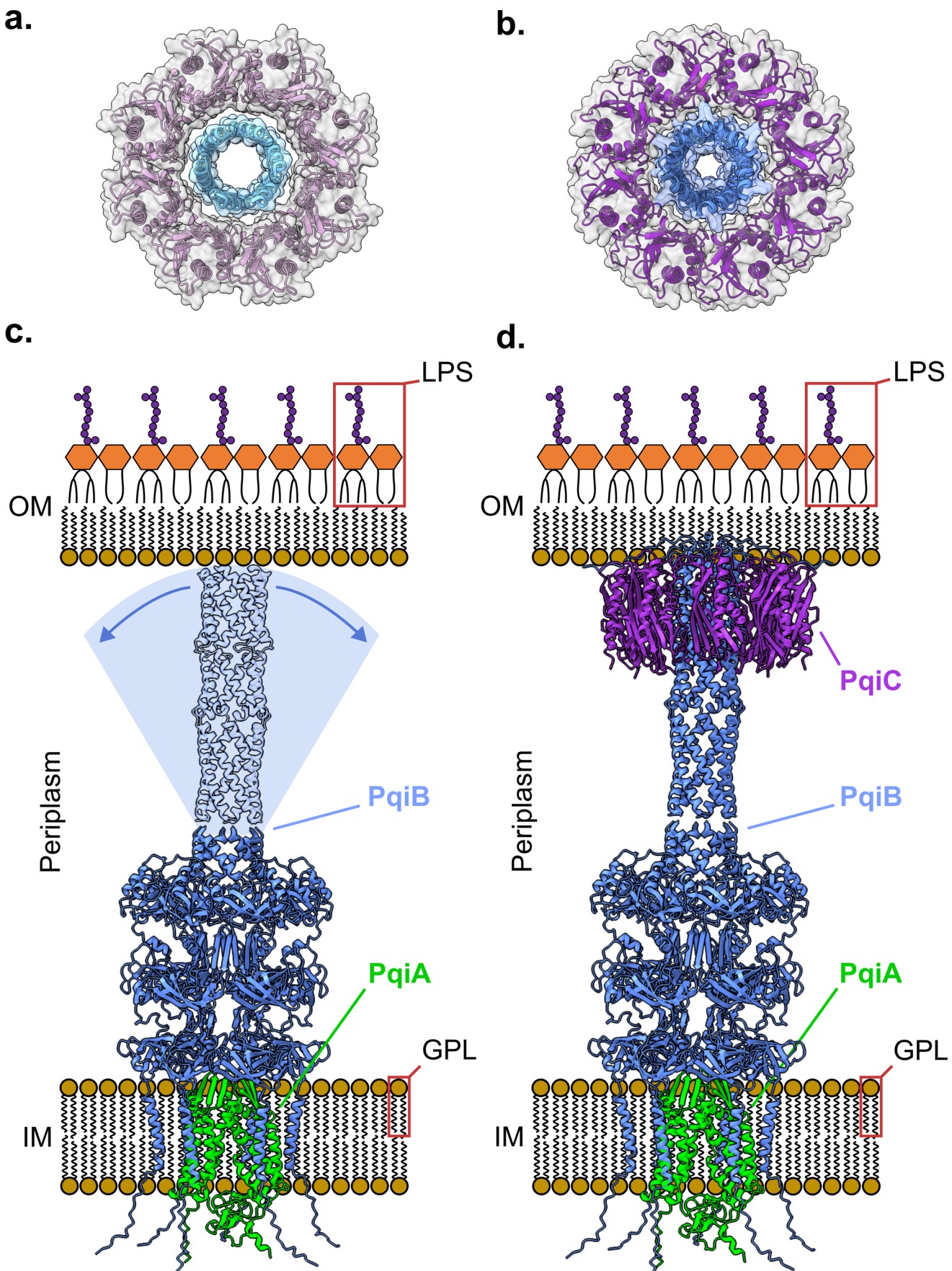

◀ **Figure 7. PqiC stabilises the interaction of PqiB with the OM.**

(A) Cartoon structures of the PqiC octamer (orchid cartoon, transparent grey surface) and the PqiB coiled-coil (sky blue cartoon and transparent blue surface) viewed from the membrane plane highlighting their potential to interface. PqiB PDB: 5UVN. (B) Cartoon depiction of the PqiB (blue cartoon, transparent blue surface) - PqiC (purple cartoon, transparent grey surface) interaction predicted by AlphaFold-Multimer, viewed from the membrane plane, indicating the complementary diameters of these two moieties. (C) Cartoon representation of the PqiAB subcomplex, as predicted by AlphaFold-Multimer, within the context of the Gram-negative cell envelope. PqiB is unable to form a stable interaction with the OM resulting in a non-functional complex. (D) Cartoon representation of the PqiABC complex, as predicted by AlphaFold-Multimer, in the context of the Gram-negative cell envelope. The presence of PqiC enables a stable interaction between PqiB and the OM allowing the formation of a functional trans-envelope complex.

unit (Amicon). Sitting drop, vapour diffusion crystallisation trials were constructed with the MemGold (Newstead et al, 2008) and MemGold 2 (Parker and Newstead, 2012) crystallisation screens (Molecular Dimensions). Crystals were grown in 96-well Combi-Clover™ 4 chamber crystallisation plates (Molecular Dimensions) containing 1 μL protein and 1 μL of a reservoir solution comprising 3 M lithium sulphate, 0.1 M MES; pH 6.5, 25% v/v PEG 400. Native diffraction data were collected at Diamond Light Source beamline I04, indexed to the I442 spacegroup and processed using the autoPROC + STARANISO pipeline (Vonrhein et al, 2018). The structure was phased via molecular replacement, performed using PHASER (McCoy et al, 2007), utilising the PqiC Alphafold predicted structure (AF-POAB10-F1), trimmed to remove the N-terminal signal peptide (residues 1–16), as the search model. The resulting model was refined using Phenix (Afonine et al, 2012) interspersed with manual inspection and adjustment in Coot (Emsley and Cowtan, 2004). The final structure comprised two copies of PqiC within the asymmetric unit, with a final $R_{work}/R_{free}$ of 0.3049/0.2469.

## Crystallisation and structure determination of PqiC$^{17-186}$

The isolated TEV cleaved PqiC$^{17-186}$ was concentrated to 15–50 mg/mL in a 10 kDa MWCO Amicon Ultra centrifugal filter unit (Amicon). Sitting drop, vapour diffusion crystallisation trials were constructed with the MemGold (Newstead et al, 2008) and MemGold 2 (Parker and Newstead, 2012) crystallisation screens (Molecular Dimensions). Crystals were grown in 96-well Combi-Clover™ 4 chamber crystallisation plates (Molecular Dimensions) containing 1 μL protein and 1 μL of a reservoir solution comprising 0.1 M sodium chloride, 0.1 M MES; pH 6.5, 33% v/v PEG 400, 4% v/v ethylene glycol. Native diffraction data were collected at Diamond Light Source beamline I04, indexed to the P61 spacegroup and processed using the autoPROC + STAR-ANISO pipeline (Vonrhein et al, 2018). The structures were phased via molecular replacement, performed using PHASER (McCoy et al, 2007), utilising chain A of the PqiC structure as the search model. The resulting model was refined using Phenix (Afonine et al, 2012) interspersed with manual inspection and adjustment in Coot (Emsley and Cowtan, 2004). The final structure comprised three copies of PqiC$^{17-186}$ within the asymmetric unit, with a final $R_{work}/R_{free}$ of 0.2041/0.2312.

## Expression and purification of PqiC-strep

DNA corresponding to PqiC residues 1–186 was chemically synthesised (Genscript) and cloned into the pET 26b plasmid, between the Nde1 and Xho1 restriction sites, to contain a C-terminal StrepTag II tag, which we refer to as PqiC-Strep

throughout (Appendix Table S5). The resulting plasmid was transformed into the E. coli C43 (DE3) cell line (Miroux and Walker, 1996) (Appendix Table S6) for expression. Cultures were grown overnight at 37 °C in Miller's lysogeny broth (Melford), supplemented with 30 μg/mL kanamycin, until an $OD_{600}$ of 0.4 at which point the temperature was reduced to 18 °C. Upon obtaining an $OD_{600}$ of 0.6, expression was induced via the addition of 1 mM IPTG and proceeded overnight at 18 °C with continual shaking (180 rpm). Cells were harvested via centrifugation at 6000 RCF for 15 min. Cell pellets were resuspended in an appropriate volume of 50 mM Tris, 150 mM NaCl; pH 8 supplemented with cOmplete™ EDTA-free protease inhibitor cocktail tablets (Roche). Cells were lysed via four passes through an Emulsiflex-C3 cell disruptor (Avestin) before cell debris was removed via centrifugation at 10,000 RCF for 10 minutes. The supernatant was decanted and spun at 165,000 RCF for 1 h to pellet the membrane fraction. Membrane pellets were resuspended in 50 mM Tris, 150 mM NaCl; 1% w/v n-dodecyl-β-D-maltoside; pH 8 at a ratio of 1 mL per 40 mg of wet membrane mass and solubilised for 3 h at 4 °C. Insoluble material was removed via centrifugation at 75,000 RCF for 30 minutes and the resulting supernatant filtered through a 0.45 μm syringe filter (Sartorius) before clarified lysate was bound overnight at 4 °C to a 5 mL StrepTrap HP column (Cytiva) pre-equilibrated in 50 mM Tris, 150 mM NaCl, 1 mM EDTA, 0.1% w/v n-dodecyl-β-D-maltoside; pH 8. The incubated column was washed with 5 CV of 50 mM Tris, 150 mM NaCl, 1 mM EDTA, 0.1% w/v n-dodecyl-β-D-maltoside; pH 8 before elution in 50 mM Tris, 150 mM NaCl, 1 mM EDTA, 2.5 mM desthiobiotin, 0.1% w/v n-dodecyl-β-D-maltoside; pH 8 (Appendix Fig. 1e). Fractions containing PqiC were pooled, concentrated and exchanged into a buffer of 50 mM Tris, 150 mM NaCl, 0.05% w/v n-dodecyl-β-D-maltoside; pH 8 via a PD-10 column (Cytiva) (Appendix Fig. S3e,f).

## PqiC-strep proteoliposome preparation

Hydrogenated or deuterated 1,2-dimyristoyl-sn-glycero-3-phos-phocholine (DMPC/dDMPC) (Avanti Polar Lipids) lipid films were resuspended in hydrogenous 20 mM Bis-Tris Propane, 150 mM NaCl, 1 mM $CaCl_2$; pH 8.5 to a final concentration of 0.1 mg/mL and sonicated until clear. The solution was supplemented with 0.02 mg/mL PqiC-Strep in 50 mM Tris, 150 mM NaCl, 0.05% w/v n-dodecyl-β-D-maltoside; pH 8.5, ensuring that the final concentration of n-dodecyl-β-D-maltoside fell below the CMC, inducing spontaneous formation of PqiC-Strep-(d/h)DMPC proteoliposomes. Protein free, control liposomes were formed by supplementing the resuspended 1,2-dimyristoyl-sn-glycero-3-phosphocholine with an identical volume of 20 mM Bis-Tris Propane, 150 mM NaCl, 0.03% w/v n-dodecyl-β-D-maltoside,

0.5 mM TCEP; pH 8.5 to that used when supplementing with 0.02 mg/mL PqiC.

## Expression and purification of PqiAB

DNA corresponding to PqiA residues 1–417 and PqiB residues 1–546 was chemically synthesised (Genscript) and cloned into the pET 26b plasmid, between the Nde1 and Xho1 restriction sites, to contain a C-terminal hexa-histidine tag, which we refer to as PqiAB throughout (Appendix Table S5). The resulting plasmid was transformed into the *E. coli* C43 (DE3) cell line (Miroux and Walker, 1996) (Appendix Table S6) for expression. Cultures were grown at 37 °C in Miller's lysogeny broth (Melford), supplemented with 30 μg/mL kanamycin, until an $OD_{600}$ of 0.4 at which point the temperature was reduced to 18 °C. Upon obtaining an $OD_{600}$ of 0.6, expression was induced via the addition of 1 mM IPTG and proceeded overnight at 18 °C with continual shaking (180 rpm). Cells were harvested via centrifugation at 6000 RCF for 15 min. Cell pellets were resuspended in an appropriate volume of 20 mM Bis-Tris Propane, 500 mM NaCl, 0.5 mM TCEP; pH 8.5 supplemented with cOmplete™ EDTA-free protease inhibitor cocktail tablets (Roche). Cells were lysed via four passes through an Emulsiflex-C3 cell disruptor (Avestin) before cell debris was removed via centrifugation at 10,000 RCF for 10 min. The supernatant was decanted and spun at 165,000 RCF for 1 h to pellet the membrane fraction. Membrane pellets were resuspended in 20 mM Bis-Tris Propane, 500 mM NaCl, 1% w/v n-dodecyl-β-D-maltoside, 0.5 mM TCEP; pH 8.5 at a ratio of 1 mL per 40 mg of wet membrane mass and solubilised for 3 h at 4 °C. Insoluble material was removed via centrifugation at 75,000 RCF for 30 min and the resulting supernatant filtered through a 0.45 μm syringe filter (Sartorius) before clarified lysate was incubated overnight at 4 °C with 1 mL of Ni-NTA Agarose (Qiagen) pre-equilibrated with 20 mM Bis-Tris Propane, 500 mM NaCl, 50 mM Imidazole, 0.03% w/v n-dodecyl-β-D-maltoside, 0.5 mM TCEP; pH 8.5. The incubated supernatant was transferred to a gravity column, washed with 20 mM Bis-Tris Propane, 500 mM NaCl, 50 mM Imidazole, 0.03% w/v n-dodecyl-β-D-maltoside, 0.5 mM TCEP; pH 8.5 before elution in 20 mM Bis-Tris Propane, 500 mM NaCl, 500 mM Imidazole, 0.03% w/v n-dodecyl-β-D-maltoside, 0.5 mM TCEP; pH 8.5. Fractions containing PqiAB were pooled, concentrated and further purified via a Superose 6 10/300 Increase column (Cytiva) equilibrated with 20 mM Tris, 150 mM NaCl, 0.03% w/v n-dodecyl-β-D-maltoside, 0.5 mM TCEP; pH 8.5 (Appendix Fig. S10).

## PqiAB proteoliposome preparation

Hydrogenated 1,2-dimyristoyl-sn-glycero-3-phosphocholine (Avanti Polar Lipids) lipid films were resuspended in hydrogenous 20 mM Bis-Tris Propane, 150 mM NaCl, pH 8.5 to a final concentration of 0.1 mg/mL and sonicated until clear. The solution was supplemented with 0.01 mg/mL PqiAB in 20 mM Bis-Tris Propane, 150 mM NaCl, 0.03% w/v n-dodecyl-β-D-maltoside, 0.5 mM TCEP; pH 8.5, ensuring that the final concentration of n-dodecyl-β-D-maltoside went below the CMC, inducing spontaneous formation of PqiAB-DMPC proteoliposomes. Protein free, control liposomes were formed by supplementing the resuspended 1,2-dimyristoyl-sn-glycero-3-phosphocholine with an identical volume of 20 mM Bis-Tris

Propane, 150 mM NaCl, 0.03% w/v n-dodecyl-β-D-maltoside, 0.5 mM TCEP; pH 8.5 to that used when supplementing with 0.01 mg/mL PqiAB.

## Quartz crystal microbalance

Quartz crystal microbalance with dissipation monitoring (QCM-D) is a surface-sensitive technique that utilises the piezoelectric effect for real-time monitoring of surface interactions.

The technique measures the resonant frequency of a quartz crystal resonator, changes in its frequency ($\Delta f$) are inversely proportional to the mass deposited on its surface, as given by the Sauerbrey equation:

$$\Delta f = \frac{2f_0^2}{A\sqrt{\rho_q \mu_q}} \Delta m \qquad (1)$$

Where $f_0$ is the resonant frequency, $A$ is the piezoelectrically excited area of the crystal and $\rho_q$ and $\mu_q$ are the density and shear modulus of the quartz crystal, respectively. QCM-D was performed using a Q-sense Analyzer QCM-D system (Biolin Scientific). Temperature and flow rate wer maintained at 25 °C and 0.1 mL/min, respectively throughout. Frequency and dissipation changes ($\Delta f$ and $\Delta D$) were monitored using multiple harmonics of the resonant frequency (n = 3, 5, 7,9, 11 & 13). Data collection was initiated in $H_2O$ before subsequent buffer exchange into 20 mM Bis-Tris Propane, 150 mM NaCl, 1 mM $CaCl_2$; pH 8.5. A suspension of PqiC-Strep/DMPC proteoliposomes (0.1 mg/mL DMPC, 0.02 mg/mL PqiC) were injected into the flow cell. Vesicle rupture was then initiated by exchange into $H_2O$ followed by subsequent return to 20 mM Bis-Tris Propane, 150 mM NaCl, 1 mM $CaCl_2$; pH 8.5, followed by exchange into 20 mM Bis-Tris Propane, 150 mM NaCl; pH 8.5. The change in osmotic gradient induced rupture of the lipid vesicles bound at the solid-liquid interface forming high quality (i.e. high coverage) supported lipid bilayers (SLBs) at the solid/liquid interface. Subsequent binding of PqiAB was monitored following the addition of a suspension of PqiAB-DMPC proteoliposomes (0.1 mg/mL DMPC, 0.0.1 mg/mL PqiAB) followed by subsequent return to 20 mM Bis-Tris Propane, 150 mM NaCl; pH 8.5.

## Neutron reflectometry measurements

Neutron Reflectometry (NR) Measurements were performed on the white beam OFFSPEC (Webster et al, 2011) reflectometer at the ISIS Neutron and Muon Source (Rutherford Appleton Laboratory, Oxfordshire, UK). Reflectivity was measured as a function of the wave vector transfer, $Q_z$ ($Q_z = (4\pi \sin \theta)/\lambda$ where $\lambda$ is wavelength and $\theta$ is the incident angle). Data were obtained at a nominal resolution (dQ/Q) of 3.5%. The reflected intensity was measured at glancing angles of 0.7° and 2.0° to give data across a wave vector transfer ($Q_z$) range of ~0.01 to 0.3 $Å^{-2}$. The total illuminated sample length was ~60 mm on all surfaces.

Details of the solid-liquid flow cell and liquid chromatography setup used in these experiments are described by us previously (Clifton et al, 2019). Briefly, solid liquid flow cells containing piranha acid cleaned silicon substrates (15 mm × 50 mm × 80 mm with one 50 mm × 80 mm surface polished to ~3 Å root mean

squared roughness) were placed onto the instrument sample position and connected to instrument controlled HPLC pumps (Knauer Smartline 1000) which enabled programmable control of the change of solution isotopic contrast in the flow cell. The samples were aligned parallel to the incoming neutron beam with the beam width and height defined using two collimating slits prior to the sample position. The sample height was aligned in such a way that the neutron beam was centred on the middle of the sample surface.

Lipid Membrane Deposition: Initially, the clean silicon substrates were characterised by NR in $D_2O$ and $H_2O$ buffer solutions. A suspension of PqiC-dDMPC (0.02 mg/mL PqiC, 0.1 mg/mL dDMPC) vesicles in the experimental buffer solution, 20 mM BTP, 150 mM NaCl, 1 mM CaCl2; pH 8.5, were injected into the cell and incubated at 25 ± 1°C for ~30 min. Non-surface bound vesicles were removed by flushing the cells with 15 mL (~5 cell volumes) of the same buffer solution before a solution of pure $H_2O$ was flushed into the cell. The change in osmotic gradient induced rupture of the lipid vesicles bound at the solid-liquid interface forming a high quality (i.e. high coverage) supported lipid bilayer (SLBs) at the solid/liquid interface. 20 mM BTP, 150 mM NaCl; pH 8.5 was introduced into the system and the resulting system characterised by NR under four solution isotopic contrast conditions, namely $D_2O$, 80% $D_2O$, Protein Matched Water (Pr-MW, 42% $D_2O$) and $H_2O$ buffer solutions. Subsequently, the addition of PqiAB-DMPC proteoliposomes (0.1 mg/mL DMPC, 0.01 mg/mL PqiAB) was performed, incubated for 30 minutes, followed by washing with 20 mM BTP, 150 mM NaCl; pH 8.5 and the surface characterised again by NR under four solution isotopic contrast conditions; $D_2O$, 80% $D_2O$, Protein Matched Water (Pr-MW, 42% $D_2O$) and $H_2O$ buffer solutions.

## Neutron reflectometry data analysis

NR data were analysed using the RasCal software (2019 version, A. Hughes, ISIS Spallation Neutron Source, Rutherford Appleton Laboratory) employing optical matrix formalism (Born and Wolf, 1999) to fit layered models of the structure across bulk interfaces. Simultaneous analysis was performed on multiple NR data sets collected under different sample and isotopic contrast conditions. Constraints were fully or partially constrained to the same surface structure in terms of thickness profile but varying in terms of neutron scattering length density (SLD). Data were fit using RasCal's custom model option where a bespoke script is used to describe the relationship between the fitted experimental parameters, the interfacial layers, the layer structure and the resulting SLD profile which was used to calculate the model reflectivity data sets which fit the experimental data.

The dDMPC-PqiC membrane structure was fit as a five-layer model, which, moving from the substrate to the bulk solution, was a thin $SiO_2$ layer, the inner bilayer head groups, two bilayer tail layers, the outer bilayer head groups and a layer of PqiC on the membrane surface. The expected molecular volume and scattering length of the bilayer head/tails were determined from the molecular components of each lipid and their expected scattering lengths. The molar ratio of head and tail components in the bilayer was

maintained by fitting an average lipid area per molecule for the bilayer where the thickness of the layers is defined by:

$$Layer\ thickness\,[\text{Å}] = \frac{component\ Molecular\ Volume\,[^3]}{Area\ per\ Molecule\,[^2]} \quad (2)$$

And the SLD was defined by:

$$SLD\,[\text{Å}^{-2}] = \frac{\sum b\,[\text{Å}]}{Component\ Molecular\ Volume\,[\text{Å}^3]} \quad (3)$$

To allow for the difference in hydration between the hydrophilic head groups and the hydrophobic tails of the bilayer two water parameters were fit, this was due to a water-associated membrane defect resulting from the differential hydration within the head and tail groups residing within the SLB plane.

Data for the complete PqiABC construct between a supported lipid bilayer of dDMPC and lower coverage hDMPC bilayer with embedded PqiA was fit using the same five-layer structure for the dDMPC-PqiC component of the interfacial structure with two additional layers added below the PqiC component. These were a thick but diffuse layer of PqiB and a layer of mixed PqiA and hDMPC. To adequately fit the experimental reflectivity profiles some mixing of the hDMPC into the dDMPC bilayer close to the silicon surface was required.

Bayesian inference of the ambiguity on the resolved structures from NR model-to-data fits and the conversion of these fits into volume fraction vs. distance profiles for the various interfacial components was conducted as described previously by us (Clifton et al, 2023).

## Design of mutant constructs utilised in phenotypic complementation assays

PqiABC mutant constructs for use within the phenotypic complementation assays were generated using the Q5 Site Directed Mutagenesis kit (New England Biolabs) as per the manufacturer's guidelines. All reactions utilised the PqiABC construct as their template, with substitutions introduced using the primers listed in Appendix Table S4.

## Phenotypic complementation assays

*E. coli* BW25113 was utilised as the wild-type (WT) parent strain throughout. A *pqiABC* knockout strain was derived from the parental strain as previously described (Datsenko and Wanner, 2000) and P1 transduced into a fresh background (Thomason et al, 2007). Finally, the *aph* cassette was removed using the pCP20 plasmid (Datsenko and Wanner, 2000). The knockout generation process was tracked via colony PCR utilising primers which bound regions flanking the *pqi* operon (Appendix Fig. 5).

Phenotypic complementation assays were performed in accordance with those previously reported for *E. coli* MCE domain proteins (Ekiert et al, 2017; Isom et al, 2020; Vieni et al, 2022). All strains were grown overnight at 37 °C in lysogeny broth, supplemented with 30 µg/mL kanamycin where appropriate. The resulting cultures were normalised to $OD_{600} = 1$ and serial diluted ($10^0$–$10^{-7}$) in fresh LB in a 96-well plate.

2.5 μL of each dilution was pipetted onto rectangular petri dishes of LB agar or LB agar supplemented with 0.25% lauryl sulfobetaine (LSB). Petri dishes were incubated overnight at 37 °C before imaging using a GelDoc system (Bio-Rad).

## Western blots of strains used in complementation assays

To confirm the expression of protein from the strains used for the complementation assays, 5 mL cultures of the *E. coli* BW25113 *ΔpqiABC* strain with each complementation plasmid, *E. coli* BW25113 *ΔpqiABC* and *E. coli* BW25113 WT strains (Appendix Table S6) were grown overnight at 37 °C in lysogeny broth, supplemented with 30 μg/mL kanamycin where appropriate. 1 mL of each culture was transferred to a 1.5 mL microcentrifuge tube and the cells pelleted by centrifugation at 4000 RCF for 15 min. Cells were resuspended in 200 uL of 2x Laemmli sample buffer (Sigma-Aldrich). Samples were then loaded onto SurePAGE™, Bis-Tris, 4–12% polyacrylamide gels (GenScript) alongside a Precision Plus Protein™ all blue prestained protein standard (BioRad). Gels were run in MES SDS running buffer (GenScript) at a constant 165 V until complete. Proteins were transferred to a nitrocellulose membrane via a Trans-Blot Turbo transfer system (BioRad) as per the manufacturer's instructions. Membranes were initially blocked in TBST supplemented with 5% (w/v) skimmed milk before overnight incubation at 4 °C with a 6xHis monoclonal antibody (Takara Bio) diluted 1:5000 with TBST supplemented with 5% (w/v) skimmed milk. Membranes were washed five times with TBST before incubation with anti-mouse HRP conjugated secondary (ThermoFisher Scientific) diluted 1:10,000 in TBST for 1 h. Membranes were washed a further five times in TBST before incubation with ECL (Cytiva) and visualisation via an Amersham 600 series imager (GE Healthcare).

## Structure prediction

The interaction between the PqiB hexamer and PqiC octamer was predicted using AlphaFold-Multimer v2.3.1 (Preprint: Evans et al, 2022; Jumper et al, 2021). Resulting predictions were visualised and analysed in ChimeraX, developed by the Resource for Biocomputing, Visualization, and Informatics at the University of California, San Francisco, with support from National Institutes of Health R01-GM129325 and the Office of Cyber Infrastructure and Computational Biology, National Institute of Allergy and Infectious Diseases (Pettersen et al, 2021).

# Data availability

The datasets and computer code produced in this study are available in the following databases: • Protein Structure: X-ray crystallographic structures. Protein Data Bank 8Q2C & 8Q2D (https://www.rcsb.org/structure/8Q2C and https://www.rcsb.org/structure/8Q2D). • Neutron reflectometry: Reflectometry binned data. Zenodo 8348250. • Modelling computer scripts: Zenodo 8348250

# Peer review information

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

## Acknowledgements

We wish to thank the funding bodies who support us, BFC, GR, HJ, RH and PW were/are jointly funded by the BBSRC and the University of Birmingham (through the Midlands Integrative Biosciences Training Partnership - Grant No. BB/M01116X/1). DJH, GWH and TJK were supported by BBSRC Research Grant No. BB/S017283/1. We thank the Science and Technology Facilities Council, the ISIS Neutron & Muon Source for allocation of beamtime (RB2300004 and RB2220703) and Diamond Light Source for access to beamline I04 under proposal number mx26803.

## Author contributions

**Benjamin F Cooper**: Conceptualization; Data curation; Formal analysis; Validation; Investigation; Visualization; Methodology; Writing—original draft; Writing—review and editing. **Giedrė Ratkevičiūtė**: Conceptualization; Data curation; Investigation; Methodology; Writing—original draft. **Luke A Clifton**: Conceptualization; Resources; Data curation; Formal analysis; Investigation; Visualization; Methodology; Writing—original draft. **Hannah Johnston**: Conceptualization; Data curation; Investigation; Methodology. **Rachel Holyfield**: Conceptualization; Data curation; Investigation; Methodology. **David J Hardy**: Investigation; Methodology. **Simon G Caulton**: Investigation; Methodology. **William Chatterton**: Investigation; Methodology. **Pooja Sridhar**: Investigation; Methodology. **Peter Wotherspoon**: Investigation; Methodology. **Gareth W Hughes**: Conceptualization; Investigation; Methodology. **Stephen CL Hall**: Conceptualization; Data curation; Software; Formal analysis; Methodology. **Andrew Lee Lovering**: Conceptualization; Formal analysis; Supervision. **Timothy J Knowles**: Conceptualization; Resources; Data curation; Software; Formal analysis; Supervision; Funding acquisition; Validation; Investigation; Visualization; Methodology; Writing—original draft; Project administration; Writing—review and editing.

## Disclosure and competing interests statement

The authors declare no competing interests.

