## [Peer Review File · EMBO Reports]

An octameric PqiC toroid stabilises the outer-membrane interaction of the PqiABC transport system.

Benjamin Cooper, Giedre Ratkeviciute, Luke Clifton, Hannah Johnston, Rachel Holyfield, David Hardy, Simon Caulton, William Chatterton, Pooja Sridhar, Peter Wotherspoon, Gareth Hughes, Stephen Hall, Andrew Lovering, and Timothy Knowles
DOI: 10.15252/embr.202357885

Corresponding author(s): Timothy Knowles (t.j.knowles@bham.ac.uk)

Review Timeline:

Submission Date:	2nd Aug 23
Editorial Decision:	12th Sep 23
Revision Received:	10th Oct 23
Editorial Decision:	26th Oct 23
Revision Received:	31st Oct 23
Accepted:	10th Nov 23

Transaction Report:

Dear Dr. Knowles

Thank you for the submission of your research manuscript to our journal. We have now received the full set of referee reports that is copied below.

As you will see, both referees consider your findings of interest and the conclusions overall well supported by the data. The referees also raise a few concerns and have suggestions how to further strengthen your manuscript, which should be addressed.

Given these constructive comments, we invite you to revise your manuscript with the understanding that the referee concerns (as detailed above and in their reports) must be fully addressed and their suggestions taken on board. Please address all referee concerns in a complete point-by-point response. Acceptance of the manuscript will depend on a positive outcome of a second round of review. It is EMBO Reports policy to allow a single round of revision only and acceptance or rejection of the manuscript will therefore depend on the completeness of your responses included in the next, final version of the manuscript.

We realize that it is difficult to revise to a specific deadline. In the interest of protecting the conceptual advance provided by the work, we recommend a revision within 3 months (December 12th). Please discuss the revision progress ahead of this time with the editor if you require more time to complete the revisions.

I am also happy to discuss the revision further via e-mail or a video call, if you wish.

Since only rather minor revisions are required, I already specify a few points below that you need to change. This will speed up the publication and editorial quality control of your revised manuscript. Further general information follows after these 7 points.

- 1) The manuscript sections are in the wrong order. Please order them like this:
Title page - Abstract - Introduction - Results - Discussion - Materials and Methods - Acknowledgements - Disclosure and competing interests statement - References - Figure legends - Tables and their legends (not EV tables) - Expanded View Figure legends
- 2) The Data Availability section goes to the end of Materials and Methods. Please specify the accession numbers for the structural data, the database, add links to the datasets and provide reviewer access (see also below).
- 3) Please update the 'Conflict of interest' paragraph to our new 'Disclosure and competing interests statement'. For more information see
<https://www.embopress.org/page/journal/14693178/authorguide#conflictsofinterest>
- 4) Please remove the Author Contributions from the manuscript file and make sure that the author contributions in our online submission system are correct and up-to-date. The information you specified in the system will be automatically retrieved and typeset into the article. You can enter additional information in the free text box provided, if you wish.
- 5) References with more than 10 authors: list only the first 10 followed by et al
- 6) The title may not exceed 100 characters incl. spaces
- 7) Supplementary Information is called "Appendix", the nomenclature is "Appendix Figure S#", "Appendix Table S#". You need a table of content with page numbers. See also below, in case you want to promote figures to the Expanded View.

- 1) a .docx formatted version of the manuscript text (including legends for main figures, EV figures and tables). Please make sure that the changes are highlighted to be clearly visible.
- 2) individual production quality figure files as .eps, .tif, .jpg (one file per figure).
Please download our Figure Preparation Guidelines (figure preparation pdf) from our Author Guidelines pages
<https://www.embopress.org/page/journal/14693178/authorguide> for more info on how to prepare your figures.
- 3) a .docx formatted letter INCLUDING the reviewers' reports and your detailed point-by-point responses to their comments. As part of the EMBO Press transparent editorial process, the point-by-point response is part of the Review Process File (RPF), which will be published alongside your paper.

4) a complete author checklist, which you can download from our author guidelines (). Please insert information in the checklist that is also reflected in the manuscript. The completed author checklist will also be part of the RPF.

5) Please note that all corresponding authors are required to supply an ORCID ID for their name upon submission of a revised manuscript (). Please find instructions on how to link your ORCID ID to your account in our manuscript tracking system in our Author guidelines ()

6) We replaced Supplementary Information with Expanded View (EV) Figures and Tables that are collapsible/expandable online. A maximum of 5 EV Figures can be typeset. EV Figures should be cited as 'Figure EV1, Figure EV2' etc... in the text and their respective legends should be included in the main text after the legends of regular figures.

7) The accession numbers and database should be listed in a formal "Data Availability " section (placed after Materials & Method) that follows the model below (see also <<https://www.embopress.org/page/journal/14693178/authorguide#dataavailability>>). Please note that the Data Availability Section is restricted to new primary data that are part of this study.

Data availability

Additional information on source data and instruction on how to label the files are available .

10) Figure legends and data quantification:

- the name of the statistical test used to generate error bars and P values,
- the number (n) of independent experiments (please specify technical or biological replicates) underlying each data point,
- the nature of the bars and error bars (s.d., s.e.m.)

- If the data are obtained from n {less than or equal to} 5, show the individual data points in addition to the SD or SEM.
- If the data are obtained from n {less than or equal to} 2, use scatter blots showing the individual data points.

11) Our journal encourages inclusion of *data citations in the reference list* to directly cite datasets that were re-used and obtained from public databases. Data citations in the article text are distinct from normal bibliographical citations and should directly link to the database records from which the data can be accessed. In the main text, data citations are formatted as follows: "Data ref: Smith et al, 2001" or "Data ref: NCBI Sequence Read Archive PRJNA342805, 2017". In the Reference list, data citations must be labeled with "[DATASET]". A data reference must provide the database name, accession number/identifiers and a resolvable link to the landing page from which the data can be accessed at the end of the reference. Further instructions are available at .

12) All Materials and Methods need to be described in the main text. We would encourage you to use 'Structured Methods', our new Materials and Methods format. According to this format, the Materials and Methods section should include a Reagents and Tools Table (listing key reagents, experimental models, software and relevant equipment and including their sources and relevant identifiers) followed by a Methods and Protocols section in which we encourage the authors to describe their methods using a step-by-step protocol format with bullet points, to facilitate the adoption of the methodologies across labs.

More information on how to adhere to this format as well as downloadable templates (.doc or .xls) for the Reagents and Tools Table can be found in our author guidelines: <

<https://www.embopress.org/page/journal/14693178/authorguide#manuscriptpreparation>>. An example of a Method paper with Structured Methods can be found here: .

13) As part of the EMBO publication's Transparent Editorial Process, EMBO Reports publishes online a Review Process File to accompany accepted manuscripts. This File will be published in conjunction with your paper and will include the referee reports, your point-by-point response and all pertinent correspondence relating to the manuscript.

Kind regards,

Referee #1:

In this manuscript, Cooper and co-workers report the crystal structure of PqiC, a lipoprotein involved in glycopospholipid trafficking to the outer membrane of gram-negative bacteria. Crytically, they show that it forms an octameric ring complex, at the periplasmic surface of the outer membrane, and that it interacts with the periplasmic component PqiB.

Overall this is a clear and well-written manuscript, that provide important new insights on our understanding of glycopospholipid trafficking. I only have a few minor comments to suggests.

- A key yet surprising observation, is that full length PqiC is an oligomer, but not the 17-187 construct, despite the first 17 Aa being disordered in the FL structure. Is there anything in the FL structure that could explain why that is? Could the presence of detergent in the FL buffer be inducing oligomerization?

- The second striking feature of the PqiC structure, is the apparent symmetry mismatch between PqiC and PqiB. The authors should include an electrostatic representation of the PqiB coiled-coil, to see if it could complement the charge of the PqiC pore. Have they tried to generate a model of the complex using AlphaFold Multimer?

- I am not well versed in the QCM-D and NR assays, so I cannot comment too much on these; however figures 3 and 4 are of

very low resolution, and some of the labels are impossible to read; this needs to be fixed.

- Lines 493-496: What is the nature of the interaction between adjacent PqiC monomers within the octamer? Is it mainly hydrophobic, or charge complementation? An additional supplementary panel might help illustrate this.

- Supplementary figure 3: on the GF traces, the authors should indicate where the void volume (V_0), and the column volume (V_c) are.

Referee #2:

The manuscript by Cooper et al describes the structure of the outer membrane lipoprotein PqiC, which functions in the bacterial Pqi lipid transport system. This lipid transport system is of broad interest because studies on the bacterial system also speak to related MCE-family complexes for lipid transport in eukaryotes. The structural analysis is of high quality, the data presentation is suitable for expert and general readership and the conclusions drawn are supported by the data. Where speculation exists around the biological impact of the work, it is made clear that it is (informed) speculation, with one exception noted in the following comments. I believe the paper would be of greater impact if more was done in terms of how PqiC interacts with PqiAB, particularly given the mis-match of symmetry that the study uncovers. I think this further impact could be achieved by further analysis of the existing data rather than a need for more experimental work. My suggestions and questions for the authors are listed below.

Q1. Could the authors move the speculative Figure 1e into the final speculation on the PqiABC structure? As it is, Figure 1 otherwise deals with data on the PqiC octamer. It is unnecessary (and potentially misleading) to place PqiB inside PqiC in this data figure.

Q2. I would suggest removing Figure 2e. Very emphatic labels are given to the sub-section layers drawn for PqiC, but the negative band has only 2 acidic residues and the positive patch is formed from a single basic residue and the distal negative patch from a single acidic residue. It seems that the subsections are arbitrarily drawn, so the placement of residues here or there along the path from the membrane surface does not appear to be significant.

Q3. The study makes use of two under-used technologies for great effect.

(i) Quartz crystal microbalance (QCM) measurements are used to establish conditions of reconstitution of the Pqi system. This has been done previously for the TAM, which may be analogous (it is not homologous) to the Pqi system in lipid transport. The QCM analysis of the TAM work is described in DOI: 10.1038/srep12905 The TAM role in lipid transport remains controversial, and therefore adds to the current excitement in the field. The Pqi system does play this role and the comparative use of QCM technology for TamAB versus PqiABC could add to the Discussion of a revised paper.

(ii) The authors also use neutron reflectometry (NR) to investigate PqiC-membrane interactions and then add to this set-up the purified PqiAB partners. Thereby assembling a trans-envelope system for the PqiABC complex that is equivalent to the previous work assembling the TamAB complex on a planar sensor surface for NR investigations. The previous TAM studies are in DOI: 10.1038/ncomms6078. NR is a hugely powerful technique - yet rarely used for membrane proteins - so adds another important aspect to the paper that could be teased out as a comparison of the systems in the Discussion of a revised paper.

Q4. For Figure 5C and 5D the contrast on some of the agar plate photographs could be improved. It is difficult to appreciate what differences are evident in the growth phenotypes of the various mutants in the data as presented.

Q5. As with the PqiC and PqiAB subcomplexes for lipid transport, a similarly perplexing symmetry mis-match has been described for Type 2 and Type 3 Secretion Systems. The outer membrane complex (i.e. secretin complex) is of distinct symmetry to the inner membrane complex that protrudes up from the periplasm. This concept and key papers have been reviewed in DOI: 10.1016/j.biochi.2022.08.019. Could this precedent be brought into the proposed modelling for how the PqiC and PqiAB complexes might interact despite their mismatch in symmetry?

Q6. A further element for comparing this work on PqiABC, the TAM and T2SS/T3SS is that a sense of cellular scale could help in distinguishing the proposed model for PqiC as a collar around PqiAB, versus PqiC as abutting to the PqiAB complex. The spatial constraints on protein complexes spanning the periplasm are becoming clear and quite precise measurements of scale across the periplasm have now been established (Reviewed in DOI: 10.1038/s41579-023-00862-w). Figure 7 could be improved if it made use of this information. This consideration would help evaluate whether PqiAB alone is sufficient to reach the outer membrane to be collared by PqiC, or whether it needs PqiC to reach out from the outer membrane in order to span the periplasm and form a conduit for lipid transfer. The existing experimental data in the current study provides the missing clues needed to make these measurements and provide for an informed discussion.

UNIVERSITY OF
BIRMINGHAM

Dr Tim Knowles
Reader in Structural Biology
Director MIBTP Birmingham

School of Biosciences
College Life and Environmental
Sciences
Vincent Drive, Edgbaston
Birmingham B15 2TT
United Kingdom

10th October 2023

Phone +44 (0)121 414 5393
Email t.j.knowles@bham.ac.uk

Dear Editor,

Thank you for the insightful comments on our manuscript provided by the two referees and for your support of its ultimate publication.

We respond to each of the reviewers' remarks on the attached pages

Yours sincerely,

Editor – comment 1

The manuscript sections are in the wrong order. Please order them like this:
Title page - Abstract - Introduction - Results - Discussion - Materials and Methods -
Acknowledgements - Disclosure and competing interests statement - References - Figure
legends - Tables and their legends (not EV tables) - Expanded View Figure legends

Authors response

We have adjusted the manuscript order accordingly.

Editor – comment 2

The Data Availability section goes to the end of Materials and Methods. Please specify the accession numbers for the structural data, the database, add links to the datasets and provide reviewer access (see also below).

Authors response

We have adjusted the manuscript accordingly.

Editor – comment 3

Please update the 'Conflict of interest' paragraph to our new 'Disclosure and competing interests statement'.

Authors response

We have adjusted the manuscript accordingly.

Editor – comment 4

Please remove the Author Contributions from the manuscript file and make sure that the author contributions in our online submission system are correct and up-to-date. The information you specified in the system will be automatically retrieved and typeset into the article. You can enter additional information in the free text box provided, if you wish.

Authors response

We have adjusted the manuscript accordingly.

Editor – comment 5

References with more than 10 authors: list only the first 10 followed by et al

Authors response

We have now addressed this in the document.

Editor – comment 6

The title may not exceed 100 characters incl. spaces

Authors response

We have adjusted the manuscript accordingly. The title now reads “**An Octameric PqiC toroid stabilises the outer-membrane interaction of the PqiABC transport system.**”

Editor – comment 7

Supplementary Information is called "Appendix", the nomenclature is "Appendix Figure S#", "Appendix Table S#". You need a table of content with page numbers. See also below, in

case you want to promote figures to the Expanded View.

Authors response

We have adjusted the text accordingly.

Reviewer#1 – Comment 1

A key yet surprising observation, is that full length PqiC is an oligomer, but not the 17-187 construct, despite the first 17 Aa being disordered in the FL structure. Is there anything in the FL structure that could explain why that is? Could the presence of detergent in the FL buffer be inducing oligomerization?

Authors response

We have adjusted the text to provide extra clarity in the differences between the constructs. The PqiC construct included a lipobox motif within its signal sequence so after processing and lipidation the mature lipoprotein sequence would begin at residue 16, with this first residue being the lipidated cysteine.

In contrast the PqiC¹⁷⁻¹⁸⁷ construct lacked any of the native signal peptide, which was instead replaced by a hexahistidine tag and TEV cleavage site. Consequently, once the purification tag was removed from the PqiC¹⁷⁻¹⁸⁷ construct, via the addition of TEV protease, the only two constructs differed only by the presence of the first cysteine residue (C16). As the only differences between constructs are the presence/absence of the N-terminally acylated cysteine we feel we have already adequately discussed the differences within the text.

The text now reads as follows

Results

“Consequently, we designed a second N-terminally truncated construct, in which residues 1-16 were replaced with a hexa-histidine tag and TEV protease recognition site, referred to as PqiC¹⁷⁻¹⁸⁷, which was expressed cytoplasmically. After TEV cleavage of the N-terminal hexa-histidine tag, the sequence of PqiC¹⁷⁻¹⁸⁷ differed from the native processed PqiC lipoprotein only by the removal of the N-terminally acylated Cysteine (C16). Nevertheless, in contrast to our PqiC construct, we observed PqiC¹⁷⁻¹⁸⁷ to elute from SEC at a volume similar to that of Ribonuclease A (13.7 kDa) (Appendix Figure S3c).”

Discussion

“We also provide the structure of a soluble PqiC¹⁷⁻¹⁸⁷ construct which differs from the mature processed PqiC lipoprotein only by the removal of the N-terminally lipidated cysteine (C16). Despite yielding a near identical monomer conformation, the PqiC¹⁷⁻¹⁸⁷ construct produced a dramatically different asymmetric unit with three copies as opposed to the two present in the native PqiC structure (Figure 1a,3a). We propose this differing arrangement results directly from the soluble or membrane/detergent associated nature of the two constructs.”

Reviewer#1 – Comment 2

The second striking feature of the PqiC structure, is the apparent symmetry mismatch between PqiC and PqiB. The authors should include an electrostatic representation of the PqiB coiled-coil, to see if it could complement the charge of the PqiC pore. Have they tried to generate a model of the complex using AlphaFold Multimer?

Authors response

Regarding the question of reciprocal electrostatics upon PqiB, we have included an additional appendix figure (Appendix Figure S5) demonstrating the electrostatics along the PqiB coiled-coil from an AlphaFold multimer prediction of the PqiB - PqiC interaction (as an experimental structure of this region remains enigmatic). We have also amended the text to include reference to the electrostatics on PqiB which now reads:

“The role of these charged regions is unclear, however, considering their positioning, it is conceivable that they may form interactions with the coiled-coil of PqiB if the two were to combine as anticipated. Nevertheless, no clear complementary charged regions are apparent towards the C-terminus of the PqiB coiled-coil within AlphaFold multimer predictions (Evans *et al*, 2022) (Appendix Figure S5) however, the lack of an experimentally determined structure means the register of residues within this region remains ambiguous.”

and

“The distinct regions of positive, negative and lipophilic residues upon this face, arising from conserved residues, are likely crucial for the correct orientation and interaction of PqiC with the membrane and require further investigation. Furthermore, AlphaFold multimer predictions (Evans *et al*, 2022) potentially imply the interaction of these charged regions with reciprocal charged regions towards the C-terminus of PqiB (Appendix Figure S5) however, the somewhat disordered nature and low confidence of the PqiB prediction throughout this region raise doubts about the validity of this interaction (Appendix Figure S7).”

Regarding the use of AlphaFold Multimer, yes we have utilised AlphaFold Multimer to predict the interaction of the PqiB hexamer and PqiC octamer. This prediction was previously utilised in the creation of the final main text figure (Figure 7) however we acknowledge that this was not properly detailed in the methods section. Consequently we have updated the methods section accordingly which now reads:

“Structure Prediction

The interaction between the PqiB hexamer and PqiC octamer was predicted using AlphaFold-Multimer v2.3.1 (Jumper *et al.*, 2012, Evans *et al.*, 2021). Resulting predictions were visualised and analysed in ChimeraX, developed by the Resource for Biocomputing, Visualization, and Informatics at the University of California, San Francisco, with support from National Institutes of Health R01-GM129325 and the Office of Cyber Infrastructure and Computational Biology, National Institute of Allergy and Infectious Diseases (Pettersen *et al.*, 2021).”

We have also included an additional Appendix figure (Appendix Figure S6) which indicates the resulting AlphaFold-Multimer prediction for the PqiB- PqiC interaction.

Reviewer#1 – Comment 3

I am not well versed in the QCM-D and NR assays, so I cannot comment too much on these; however figures 3 and 4 are of very low resolution, and some of the labels are impossible to read; this needs to be fixed.

Authors response

Apologies for this, it appears this was an issue with conversion to pdf that we missed. We have rectified this by supplying high resolution images separately and reworking the figures to improve visibility of the labels.

Reviewer#1 – Comment 4

Lines 493-496: What is the nature of the interaction between adjacent PqiC monomers within the octamer? Is it mainly hydrophobic, or charge complementation? An additional supplementary panel might help illustrate this.

Authors response

We have updated the text to include more information about the nature of the interaction between the PqiC monomers. Furthermore we have also included an additional Appendix Figure (Appendix Figure S4) which illustrates the intermolecular interactions as well as an additional Appendix Table (Appendix Table S2) which details the PqiC interface interactions identified by PDBePISA.

The text now reads as follows:

“We utilised the PDBePISA server (Krissinel & Henrick, 2007) to investigate the intermolecular interface in more detail. PISA revealed the interface to be stabilised by 12 intermolecular hydrogen bonds and a single intermolecular salt bridge between K166 and D68 (Appendix Figure S4, Table S2). Furthermore, the complex formation significance score (CSS) of the interface was reported at 1, indicating the interface to play an essential role in complex formation.”

Reviewer#1 – Comment 5

Supplementary figure 3: on the GF traces, the authors should indicate where the void volume (V_o), and the column volume (V_c) are.

Authors response

We have updated the figure accordingly to now include the void volume and column volumes.

Reviewer#2 – Comment 1

Could the authors move the speculative Figure 1e into the final speculation on the PqiABC structure? As it is, Figure 1 otherwise deals with data on the PqiC octamer. It is unnecessary (and potentially misleading) to place PqiB inside PqiC in this data figure.

Authors response

As requested, we have moved Figure 1e into the final speculation figure on PqiABC.

Reviewer#2 – Comment 2

I would suggest removing Figure 2e. Very emphatic labels are given to the sub-section layers drawn for PqiC, but the negative band has only 2 acidic residues and the positive patch is formed from a single basic residue and the distal negative patch from a single acidic residue. It seems that the subsections are arbitrarily drawn, so the placement of residues here or there along the path from the membrane surface does not appear to be significant.

Authors response

In this case we disagree, we feel that the figure more clearly highlights the residues of interest in this region making it easier for the reader to follow. Although the reviewer highlights that the negative band has only 2 acidic residues, and this is true but as part of the whole octameric ring the negative band has 16 residues, surrounding the toroid, and this is what we were attempting to show. We therefore feel that the placement of these residues along the path from the membrane surface is highly relevant to the narrative.

Reviewer#2 – Comment 3

The study makes use of two under-used technologies for great effect.

(i) Quartz crystal microbalance (QCM) measurements are used to establish conditions of reconstitution of the Pqi system. This has been done previously for the TAM, which may be analogous (it is not homologous) to the Pqi system in lipid transport. The QCM analysis of the TAM work is described in DOI: 10.1038/srep12905 The TAM role in lipid transport remains controversial, and therefore adds to the current excitement in the field. The Pqi system does play this role and the comparative use of QCM technology for TamAB versus PqiABC could add to the Discussion of a revised paper.

(ii) The authors also use neutron reflectometry (NR) to investigate PqiC-membrane interactions and then add to this set-up the purified PqiAB partners. Thereby assembling a trans-envelope system for the PqiABC complex that is equivalent to the previous work assembling the TamAB complex on a planar sensor surface for NR investigations. The previous TAM studies are in DOI: 10.1038/ncomms6078. NR is a hugely powerful technique - yet rarely used for membrane proteins - so adds another important aspect to the paper that could be teased out as a comparison of the systems in the Discussion of a revised paper.

Authors response

We have adjusted the manuscript to reference the two papers advised by the reviewer here detailing the use of planar surface techniques to study the TamAB interaction. The relevant section of the discussion now reads:

“The distance between the two reconstituted membranes in our PqiABC NR system was calculated to be 28.1 ± 1.4 nm. This value is congruous with the upper limit of the 21-27 nm range for the measured width of the periplasm *in vivo* (Mandela *et al*, 2022; Matias *et al*, 2003). This is much larger than the 200 Å estimate of the periplasmic span during assembly of the translocation and assembly module upon planar surfaces using both QCM-D and NR (Selkrig *et al*, 2015; Shen *et al*, 2014). Here the system comprised the Omp85 outer membrane protein TamA and the AsmA-like protein TamB. However, this work differs from

our own in one fundamental aspect. Here, the use of a TamB construct lacking its N-terminal TM region prevented the formation of a second distal membrane. In contrast, the use of native PqiB and PqiA constructs within our own studies enabled the sequential assembly of the complete trans-membrane Pqi complex with a double membrane system.”

Reviewer#2 – Comment 4

For Figure 5C and 5D the contrast on some of the agar plate photographs could be improved. It is difficult to appreciate what differences are evident in the growth phenotypes of the various mutants in the data as presented.

Authors response

We agree and have re-made the figure to aid clarity by ensuring that the quality of the images remains high and also adjusted the contrast of the images slightly to aid interpretation.

Reviewer#2 – Comment 5

As with the PqiC and PqiAB subcomplexes for lipid transport, a similarly perplexing symmetry mis-match has been described for Type 2 and Type 3 Secretion Systems. The outer membrane complex (i.e. secretin complex) is of distinct symmetry to the inner membrane complex that protrudes up from the periplasm. This concept and key papers have been reviewed in DOI: 10.1016/j.biochi.2022.08.019. Could this precedent be brought into the proposed modelling for how the PqiC and PqiAB complexes might interact despite their mismatch in symmetry?

Authors response

We thank the reviewer for bringing this to our attention and have subsequently updated the manuscript to include reference to these systems in our discussion. The text now reads:

“Symmetry mismatch has also been observed in other trans-envelope systems including both the Type 2 (T2SS) and Type 3 (T3SS) Secretion Systems. Here, 15:12 and 15:24 stoichiometries have been observed between the C-modules and their IM partners, respectively (Chemyatina & Low, 2019; Hu *et al*, 2018). Exactly how this symmetry mismatch might be overcome remains unclear, but there is mounting evidence to suggest the overall stoichiometry of the secretin may fluctuate to be compatible with the stoichiometry of its IM partner (Barbat *et al*, 2023; Hay *et al*, 2017). In principle the same could occur for PqiC, however, this seems somewhat less likely due to the coinciding diameters of the hexameric PqiB coiled-coil and octameric PqiC pore.”

Reviewer#2 – Comment 6

A further element for comparing this work on PqiABC, the TAM and T2SS/T3SS is that a sense of cellular scale could help in distinguishing the proposed model for PqiC as a collar around PqiAB, versus PqiC as abutting to the PqiAB complex. The spatial constraints on protein complexes spanning the periplasm are becoming clear and quite precise measurements of scale across the periplasm have now been established (Reviewed in DOI: 10.1038/s41579-023-00862-w). Figure 7 could be improved if it made use of this information. This consideration would help evaluate whether PqiAB alone is sufficient to reach the outer membrane to be collared by PqiC, or whether it needs PqiC to reach out from

the outer membrane in order to span the periplasm and form a conduit for lipid transfer. The existing experimental data in the current study provides the missing clues needed to make these measurements and provide for an informed discussion.

Authors response

We thank the reviewer for bringing this reference to our attention and have subsequently updated the manuscript to include discussion about the scale of our proposed complex and how this relates to that of the periplasm. The text now reads:

“Previous cryo-EM studies have reported the periplasmic portion of PqiB to be ~230 Å in length (Ekiert *et al*, 2017), whilst our own NR studies reveal PqiC to sit ~50 Å proud of the outer-membrane surface. Consequently, the observed distance may be accounted for simply by the stacking of these two components and thus insertion of the PqiB coiled-coil into the PqiC toroid is not required. Nevertheless, we anticipate the interaction between PqiB and PqiC to require the PqiB coiled-coil to locate into the centre of the PqiC octamer, a notion further supported by the AlphaFold multimer prediction of the PqiABC complex (Evans *et al*, 2022) (Figure 7) (Appendix Figure S5,7). Thus, assuming the insertion of PqiB within PqiC, the marginally larger than anticipated intermembrane distance measured may simply reflect the inherent low resolution *in vitro* nature of our NR study. It must also be noted that our NR experiments were conducted with the flow cell located below the silicon substrate, thus the assembled system, with the PqiAB-liposome tethered, was suspended beneath the substrate and would therefore be subject to the effects of gravity.”

More precise evaluation of the ability of PqiB to reach the outer membrane itself is limited by the lack of an experimentally derived structure for the entirety of the protein. Our current model utilises an AlphaFold predicted structure of PqiB however unfortunately the C-terminus is still poorly predicted and thus we are unable to determine the full extent of the protein. Indeed the recent publication of an endogenous Mtb MCE transporter raises the possibility of more complex C-terminal domains that AF is unable to currently predict. For this reason we feel uncomfortable in potentially overanalysing the interaction of PqiB with the OM given the current lack of an experimentally derived structure.

Dear Dr. Knowles

Thank you for the submission of your revised manuscript to EMBO reports. I have assessed your revision and your response to the minor concerns raised by the referees, which seemed adequate but before I can formally accept your manuscript for publication, I need you to address some minor points below:

- Please provide up to 5 keywords
- Please add callouts for Appendix Table S5 and S6 in the text, where appropriate.
- Source data for Figure 5C and 5D needs to be clearly labeled and it needs to be sorted into subfolders. In this case you make a folder for Figure 5 and in this folder you place the source data for C into a folder called 5C and the one for D into a folder called 5D. If necessary, you can also add a README.txt file. When submitting replicate data as source data please add a subfolder in the relevant source data figure/panel folder, name the file the same as the data shown in the figure and add the term 'replicate'.
- Neutron reflectometry data on Zenodo: Will it be obvious to readers to which sections in the manuscript the deposited source data refers to? There seems to be no further description, but maybe it is obvious for experts.
- Please rename the "Methods" section to "Materials and Methods"
- Title: should "Octameric" be in lower case letters?
- Please remove the following statement from the Data availability section: "In addition, the data that support the findings of this study are available from the corresponding author on request." Text in this section should only refer to data deposited in databases.
- Please add links that directly resolve to the datasets deposited on Protein Data Bank.
- Our production/data editors have asked you to clarify several points in the figure legends (see below). Please incorporate these changes in the manuscript and return the revised file with tracked changes with your final manuscript submission.
 - 1) Please note that the error bars are not defined in the legend of figures 4b, f.
 - 2) Please note that information related to n is missing in the legend of figures 4b, f.
- Evans et al 2022 is a preprint. You correctly labeled it with [PREPRINT] in the reference list but it also needs a label in the text as follows: (preprint: Evans et al, 2022).
- In the methods section on Neutron Reflectometry Data Analysis (in particular from line 842 to line 869) the description is very similar to that in Luke Clifton et al, Science Advances 2023, Supplementary Materials (DOI: 10.1126/sciadv.adg794). While it is fine to use a similar description of methodology used in a previous publication - no need to reinvent the wheel -, I nevertheless suggest citing this manuscript in the respective methods section, just as you did in line 782 with the reference to Clifton et al, 2019.
- Finally, EMBO Reports papers are accompanied online by A) a short (1-2 sentences) summary of the findings and their significance, B) 2-3 bullet points highlighting key results and C) a synopsis image that is 550x300-600 pixels large (width x height) in PNG for JPG format. You can either show a model or key data in the synopsis image. Please note that the size is rather small and that text needs to be readable at the final size. Please send us this information along with the revised manuscript.
- On a different note, I would like to alert you that EMBO Press offers a new format for a video-synopsis of work published with us, which essentially is a short, author-generated film explaining the core findings in hand drawings, and, as we believe, can be very useful to increase visibility of the work. This has proven to offer a nice opportunity for exposure i.p. for the first author(s) of the study.

Please see the following link for representative examples and their integration into the article web page:

<https://www.embopress.org/doi/full/10.15252/emj.2019103932>

With kind regards,

The authors have addressed all minor editorial requests.

Dr. Timothy Knowles
University of Birmingham
School of Biosciences
Vincent Drive
Edgbaston
Birmingham, West Midlands B15 2TT
United Kingdom

Dear Tim,

I am very pleased to accept your manuscript for publication in the next available issue of EMBO reports. Thank you for your contribution to our journal.

Kind regards,

Martina
